# Mechanistic insights into steroid hormone-mediated regulation of the androgen receptor gene

**Andrew D. Gillen[1][¤], Irene Hunter[1], Ekkehard Ullner[2], Iain J. McEwan[1]***

**1** Institute of Medical Sciences, School of Medicine, Medical Sciences and Nutrition, University of Aberdeen, Scotland, United Kingdom, **2** Department of Physics, Institute of Complex Sciences and Mathematical Biology University of Aberdeen, Scotland, United Kingdom

¤ Current address: School of Molecular Biosciences, University of Glasgow, Glasgow, United Kingdom
* iain.mcewan@abdn.ac.uk

## Abstract

Expression of the androgen receptor is key to the response of cells and tissues to androgenic steroids, such as testosterone or dihydrotestosterone, as well as impacting the benefit of hormone-dependent therapies for endocrine diseases and hormone-dependent cancers. However, the mechanisms controlling androgen receptor expression are not fully understood, limiting our ability to effectively promote or inhibit androgenic signalling therapeutically. An autoregulatory loop has been described in which androgen receptor may repress its own expression in the presence of hormone, although the molecular mechanisms are not fully understood. In this work, we elucidate the mechanisms of autoregulation and demonstrate, for the first time, that a similar repression of the *AR* gene is facilitated by the progesterone receptor. We show that the progesterone receptor, like the androgen receptor binds to response elements within the *AR* gene to effect transcriptional repression in response to hormone treatment. Mechanistically, this repression involves hormone-dependent histone deacetylation within the *AR* 5'UTR region and looping between sequences in intron 2 and the transcription start site (TSS). This novel pathway controlling *AR* expression in response to hormone stimulation may have important implications for understanding cell or tissue selective receptor signalling.

## 1. Introduction

The sex hormone testosterone plays multiple roles in both reproductive and peripheral tissues in both men and woman. It also acts as a pro-hormone for the more potent androgen dihydrotestosterone (DHT) and the principal oestrogenic steroid hormone, oestradiol (E2) [1]. Testosterone and DHT both work through the same protein, the androgen receptor (AR) which is encoded by the *AR* gene located on the X-chromosome [2]. The *AR* gene is expressed ubiquitously in human tissues although the levels of the mRNA fluctuate widely.

Interestingly, both positive and negative feedback loops have been proposed to regulate AR mRNA in target tissues. Indeed, negative autoregulation is widely supported in prostate cancer

**Data Availability Statement:** All relevant data are within the paper and its Supporting information files.

**Funding:** This work was supported by small project grants from NHS Grampian (17/040) and Friends

of Anchor (RS 2019 001). ADG was supported by a University of Aberdeen PhD studentship, warded competitively by the Institute of Medical Sciences. Funding for open access charge: None. There was no additional external funding received in this study. The funders had no role in study design, data collection and analysis, decision to publish, or preparation of the manuscript.

**Competing interests:** The authors have no conflicts of interest to declare.

models and cell lines which endogenously express the receptor [3, 4]. This autoregulation displays a high degree of tissue specificity, as breast cancer cells show similar autorepression [5, 6]; whilst bone and postmenopausal endometrium display an inverted AR-mediated upregulation of AR transcription [6–8]. Even within prostate cells, androgenic AR control is nuanced, as studies in cells which do not express endogenous AR demonstrate DHT-mediated upregulation of transfected AR cDNA constructs [9, 10].

Expression of the *AR* gene is driven from a TATA-less promoter and involves multiple Sp1 binding regions (see [11] and references therein). Auto-upregulation of AR has been associated with AR binding, together with c-myc, to a 350bp DNA fragment spanning AR exons 4 and 5 [10]. A putative site upstream of the transcription start site, that facilitates ERG/AR co-regulation of AR expression has been described, although a precise definition of this element is unknown and evidence relating this to hormone-mediated repression is lacking [12]. More recently, an enhancer region, around 625 to 650 kb upstream, has also been identified [13, 14]. Interestingly, the evidence suggests this may be developmentally important and is reactivated in cases of advanced prostate cancer. Notably, binding sites for transcription factors associated with both positive and negative regulation were found in this region, such as FOXA1, HOXB13 and GATA2 [13].

Detailed studies from the Balk group have identified sequences bound by the AR within intron 2 of the *AR* gene, repressing expression via LSD1 demethylation of H3K4. This region is poorly defined, with multiple putative AR response element half-sites identified [15, 16]. A further AR response element associated with gene repression was identified in the AR 5'UTR, which possesses a more defined DNA response element sequence [17]. In this case, while no clear mechanism for repression was uncovered, LSD1 was not thought to bind the 5'UTR element.

The intron 2 element contains half site motifs, 5'-AGAACA-3' [16], whilst the 5' UTR element contains a partially palindromic, repeat of the motif, 5'-AGAACCctcTGTTT-3' [17]. In this latter orientation in particular, this motif also presents a putative binding site for other hormone receptors including the glucocorticoid receptor (GR), the mineralocorticoid receptor (MR) and the progesterone receptor (PR) [18]. These receptors, along with the AR, are defined as 3-keto-steroid receptors, and are very closely related, albeit with PR (expressed as two isoforms, PR-A and PR-B) more similar to the AR whilst GR and MR share the greatest degree of homology.

There is a growing body of evidence that other steroid receptors can influence AR signalling. GR has been implicated as a clinically relevant AR substitute in advanced prostate cancer, recapitulating AR signalling [19, 20]. Oestrogen activity has been shown to promote AR expression in female uterine tissues [8, 21], whilst in mammary and prostate cells the ERs, particularly ERβ, have been shown to repress AR [5, 22]. PR has demonstrated repression of AR within both the endometrium [23, 24] and mammary cancer cells [5]. In the prostate, the role of PR is as-yet unclear, with reports of both tumour suppressive roles [25, 26] and a role in disease progression and clinical failure [27, 28].

Given the importance of testosterone and the androgen receptor in both normal physiology as well as pathological conditions it is important to more fully understand the molecular mechanisms underpinning regulation of the *AR* gene. We have investigated the roles of different steroid hormone receptors in controlling AR expression using prostate cancer cell models, with a focus on known genomic loci. We have now identified a novel progesterone-regulated pathway repressing AR expression in prostate epithelia. Mechanistic analysis delineates, for the first time, that both AR and PR bind to sites within the 5' UTR and Intron 2 of the *AR* gene, exerting similar, but distinct effects on overall gene expression. Specifically, we show that both receptors leverage histone deacetylation machinery to facilitate hormone-mediated AR repression.

## 2. Material and methods

### 2.1 Cell culture

Four prostate cell lines were used: LNCaP, VCaP, and PC3 (all from ATCC) and 22rv1 (gift from Prof. R. Narayanan, University of Tennessee Health Science Centre). VCaP cells were grown in DMEM, while LNCaP, 22rv1 and PC3 were grown in RPMI-1640. Cells were grown without antibiotics at 37˚C in a humidified atmosphere containing 95% air and 5% CO2. All growth media was supplemented with 10% foetal bovine serum. Cells were transferred to media supplemented with 5% charcoal-stripped foetal bovine serum 24 hours prior to treatment. All steroid receptor ligands (dexamethasone (StressMarq Biosciences, Victoria, Canada), 17β-oestradiol (Sigma-Aldrich, Darmstadt, Germany), diarylpropionitrile (Abcam, Cambridge, England), dihydrotestosterone (Sigma-Aldrich, Darmstadt, Germany) and progesterone (Sigma-Aldrich, Darmstadt, Germany)) were used at a final concentration of 10 nM. Actinomycin D (Sigma-Aldrich, Darmstadt, Germany) was used at a concentration of 10 μM. HDAC inhibitors (Sigma-Aldrich, Darmstadt, Germany), SAHA and Sodium Butyrate were used at 50 nM and 10 μM respectively.

### 2.2 Transfection

Transient plasmid transfection was carried out using JetPEI polyethyleneimine (Polyplus-transfection, France) transfection reagent, according to manufactures' protocol, with cells at approximately 70% confluence. Treatments were applied 24 hours following transfection.

### 2.3 Western blotting

Proteins were extracted from cells with RIPA buffer (20 nM Tris-HCl, 150 nM NaCl, 1% Igepal, 0.5% Sodium Deoxycholate and 0.1% SDS). 25 μg protein was fractionated by SDS-polyacrylamide gel electrophoresis, transferred to PDVF membranes, probed with specific antibodies and visualised by enhanced chemiluminescence using a LI-COR scanner. A list of antibodies used is provided in S1 Table.

### 2.4 Quantitative reverse transcriptase PCR

VCaP cells were treated with hormone and RNA was isolated using the RNeasy mini kit (Qiagen, Hilden, Germany), and genomic DNA digested. The quantity and purity (A260/A280) of each sample was determined using a nanodrop spectrophotometer and 25 ng RNA was used for each reaction.

Quantitative Reverse Transcriptase PCR (qRT-PCR) was carried out using Precision PLUS OneStep RT-qPCR Master Mix (Primer design, Camberley, England) plus SYBR green (Qiagen, Hilden, Germany) and ROX. All primers were used at a final concentration of 6 pmol (Final reaction volume 20 μL).

qRT-PCR target sequences were AR (Full length and selected AR-vs), Accession NM_000044.6 (Amplicon length 103 bp) and GAPDH, Accession NM_002046.7 (Amplicon length 66 bp). In all cases primers spanned exon junctions. GAPDH was chosen as a reference gene to align with previous RNA quantification [17]. Primers used are listed in S2 Table.

Cycling conditions were altered from manufacturer's guidelines to avoid interference of primer dimerization with quantification [29]. A StepOnePlus Thermocycler was utilised to carry out qPCR. Immediately prior to amplification, samples were reverse transcribed at 55˚C for 10 min, then enzyme was activated at 95˚C for 5 min. The following cycling program was used: 95˚C, 10 s (denaturation); 60˚C, 4 s (annealing); 72˚C, 32 s (extension); 80˚C, 5 s (Primer dimer denaturation and data collection). This cycle was repeated 35 times.

Each qRT-PCR run included a melt-curve analysis, to exclude formation of undesired products, along with a serial RNA dilution to ensure acceptable (90–110%) efficiency. Non-specific amplification was controlled for using non-template controls (NTCs) on every qPCR run. These were deemed acceptable if NTC Cqs were at least 10 higher than sample Cq values. Amplification of gDNA was excluded by using a no-RT control.

qRT-PCR data analysis used StepOne software (Version 2.3, available from Thermo Scientific). Cycle thresholds were automatically set for each study. Outliers were automatically highlighted by the program and excluded based on: lack of amplification; bad passive reference (ROX) signal; or significant Cq deviation from replicates, based on a modified Grubb's test. In all cases, qRT-PCR samples were run in technical triplicates and three independent biological replicates were analysed for each condition. Statistical analysis utilised non-normalised (mean $\Delta$Cq) values from the three biological replicates.

## 2.5 ChIP-qPCR

VCaP cells were transfected with pCMV-hPR-A/B, then treated with vehicle (EtOH) or 10 nM hormone (DHT or progesterone) for 4 hours. Cells were fixed in 1% formaldehyde for 10 minutes at 37˚C. Chromatin was harvested in the presence of SIGMAFast protease inhibitors (Sigma-Aldrich, Darmstadt, Germany) and sonicated using a Soniprep 150 probe sonicator to a size of 200–600 bp (15x 30 s pulses, 5 amplitude microns). Input samples (0.83% total chromatin) were saved for qPCR normalisation.

Samples were pre-cleared and immunoprecipitated overnight in appropriate antibody or IgG control (list in S1 Table). Immunoprecipitated fragments were captured by incubating with a mixture of Protein A and Protein G Dynabeads. After washing, DNA eluted from the beads was treated with RNase A and Proteinase K and purified using a QIAquick Purification Kit (Qiagen, Hilden, Germany).

Recruitment of the AR or PR to response elements in DNA samples was quantified via qPCR using RT2 SYBR Green qPCR Mastermix. All primers were used at a final concentration of 6 pmol. 5 μL ChIP DNA samples were used undiluted for qPCR, to preserve differences in DNA concentration (Final reaction volume 20 μL).

ChIP-qPCR target sequences can both be found within the AR gene, Accession NM_000044.6: the 5' UTR element and Intron 2 element amplicons, size 221 bp and 399 bp respectively. Primers used are listed in S2 Table.

An initial denaturation was run at 95˚C for 10 minutes, followed by the cycling program: 95˚C, 15 s; 60˚C, 5 s; 72˚C, 40s. qPCR run for 40 cycles. Input samples used for normalisation of results, and to produce standard curves (runs accepted between 90–110% efficiency). Internal qPCR controls (Melt curves, NTCs) were used as described for qRT-PCR.

Data analysis was carried out using StepOne software, with the same parameters for Cq determination and sample exclusion. Samples were run in technical triplicate, with each experiment repeated independently three times. Statistical analysis utilised quantification of each sample as a percentage of the equivalent input, to account for differences in chromatin harvesting.

## 2.6 Transactivation assay

LNCaP and PC3 cells were transfected with phAR1.6Luc alone, or phAR1.6Luc (Hay et al 2014) plus CMV-hPR-A or CMV-hPR-B respectively 24 hours prior to hormone treatment. Cells were treated with 10 nM hormone for 24 hours and protein was extracted using Passive Lysis Buffer (Promega, Wisconsin, U.S.). Cell debris was removed by centrifugation, and 10 μL of each lysate was loaded in triplicate to a 96-well plate. Luciferase activity in each sample was

measured using a GloMax 96 Microplate luminometer (Promega) by injecting a mixture of luciferase buffer (12 mM MgSO47H2O, 30 mM GlyGly, 1.5 mM Na2ATP) plus 11 μM luciferin (Invitrogen). Relative light unit readings were normalised for total protein concentration. Experiments were run in biological triplicate and statistical analysis was carried out using luciferase activity/mg/mL protein values.

## 2.7 ChIP-Loop

VCaP cells were transfected, with pCMV-hPR-A/B, and treated with vehicle (EtOH) or 10 nM hormone (DHT or progesterone) for 4 hours. Cells were fixed in 1% formaldehyde for 10 minutes at 37˚C. Chromatin was harvested in the presence of protease inhibitors. The chromatin pellet was resuspended in Buffer A (10 mM Hepes (pH 7.9), 10 mM KCl, 1.5 mM MgCl2) and allowed to swell on ice. Pellets were isolated, homogenised using a glass homogeniser and nuclei resuspended in NEbuffer 2.1 (New England Biolabs, Massachusetts, U.S.), supplemented with SIGMAFast protease inhibitors. 400 units of Bst*YI* (New England Biolabs, Massachusetts, U.S.) was added to each sample, and digestion was carried out overnight at 37ºC. Bst*YI* was inactivated, samples were separated from cell debris and supplemented with ChIP dilution buffer (1.1% Triton X-100, 0.01% SDS, 1.2 mM EDTA, 16.7 mM Tris-HCl (pH 8.1), 167 mM NaCl). Samples were pre-cleared then immunoprecipitated overnight in appropriate antibody (list in S1 Table).

Immunoprecipitated fragments were captured by incubating with a mixture of Protein A and Protein G Dynabeads (Thermo Scientific, Massachusetts, U.S). Supernatant was discarded and beads were resuspended in DNase-free H2O. On-bead ligation was carried out at 4 ºC overnight by addition of Ligation cocktail (1% Triton X-100, 1% Ligation Buffer, 8 mg BSA, 1.05 mM ATP) and 25 Weiss units of T4 Ligase (Thermo Scientific, Massachusetts, U.S.). Following ligation, T4 ligase was inactivated, and bead-bound DNA was washed and eluted.

A traditional PCR-based approach, utilising GoTaq DNA polymerase (Promega, Wisconsin, U.S.) was employed to analyse which ligation products were present. Looping from the AR TSS to either its Intron 2 repressor or an arbitrary upstream control region was measured in a PCR reaction with one AR TSS primer, and one primer in the remote site of interest. Internal primers were used as a control, to show presence of DNA in all conditions. This PCR used a 2 min initial denaturation at 95˚C, then cycled: 95˚C, 30 s; 60˚C 30s; 72˚C 30s. 40 cycles were performed with a final extension for 5 mins at 72˚C. As a positive control, artificial chromosomes were used to prepare a control ligation library covering the region of interest. An NTC was used to control for non-specific amplification. PCR products were analysed on a 2% agarose gel. Primers used are listed in S2 Table.

## 2.8 ChIP-seq data set analysis

To expand our studies into other steroid hormone-dependent tissues, we identified publicly available AR and PR ChIP-seq datasets from breast and endometrial tissue. Using the GEO (Gene Expression Omnibus) service provided by NCBI, we identified the following datasets: AR ChIP-seq in endometrial stroma from polycystic ovarian syndrome patients [30]; AR ChIP-seq in breast cancer cell lines, including with hormone stimulation [31]; isoform-specific PR ChIP-seq in a normal endometrial stromal cell line [32]; PR ChIP-seq in patient samples from the endometrial cancer leiomyoma [33]; and PR ChIP-seq in response to agonist treatment in a breast cancer cell line [34].

To ensure findings were unbiased by data processing, we elected to re-analyse all datasets from raw FASTA files. FASTAs were downloaded to the online bioinformatics platform Galaxy for processing [35]. Raw reads were aligned to the human genome (hg19 build) using

HISAT2 [36] using default settings. To analyse statistically significant differences between read depth at specific loci between ChIP-seq experiments, we compared aligned read data using MACS2 to call enriched peaks [37].

### 2.9 Statistical analysis

Statistical analyses were carried out using IBM SPSS Statistics (Version 25). In all cases, statistical analyses were carried out using results from three independent biological replicates. In all cases, statistics were carried out using un-normalised data: only corrected band intensity for Western Blot analysis, and Delta Ct for qPCR analysis. Unpaired student's t-test analysis was used to make individual comparisons, whilst comparisons across multiple groups utilised ANOVA tests, with post-hoc analysis using Tukey's test. Resulting two-tailed p-values determined to be significant ($p < 0.05$) are noted throughout (* = $p < 0.05$; ** = $p < 0.01$; *** = $p < 0.001$). Specific p values are given for relevant comparisons. Data is presented as mean values from biological replicates, ± standard error of the mean.

## 3. Results

### 3.1 Different steroid hormones repress prostate cell AR expression

To investigate hormone-mediated changes in AR mRNA, VCaP cells were treated with 10 nM dihydrotestosterone (DHT), dexamethasone (Dex), 17β-oestradiol (E2) or progesterone (P4), to activate the cognate steroid hormone receptors. VCaP cells were chosen for these studies as they overexpress wild-type full-length AR as well as several splice variants lacking the hormone binding domain and the GR [38]. However, VCaP cells do not endogenously express either ERα or PR (A or B) (S1 Fig) and therefore cells were transfected with expression plasmids for these receptors (pCDNA3-ERα, pCMV-hPR-A or pCMV-hPR-B) prior to hormone treatment. It should be noted that transfection with CMV-PR-B did not induce expression of the PR-B isoform exclusively, due to the alternate start site for PR-A present in its coding region. Efficiency of transfection was confirmed via Western blot, demonstrating appropriate ERα, PR-A and PR-B expression (S2A and S2B Fig). ERβ status of cell lines could not be reliably determined due to a paucity of reliable antibodies [39]. To ensure that the receptor was present, cells were transfected with pDC315-ERβ-CFP prior to analysis of the effects of ERβ. Efficacy of this transfection was confirmed by blotting against the CFP tag fused to the expressed ERβ (S2 Fig).

Analysis of transcript levels confirmed that DHT treatment significantly ($p = 0.004$) reduced AR mRNA levels relative to vehicle-treated controls (Fig 1A), while Dex mediated activation of glucocorticoid receptor had no significant effect. Similarly, E2 treatment in the presence of ERα elicited no significant response in terms of mRNA levels (Fig 1B). By contrast, activation of ERβ with either E2 or selective ligand DPN resulted in a striking reduction in AR levels (Fig 1C). Notably, however, neither ligand reduced AR expression to a level significantly lower than ERβ transfection alone (S3 Fig). This raises the possibility that ERβ-mediated reduction in AR transcript levels may, at least in part, be ligand-independent, although a further modest decrease in expression of both AR mRNA and protein was observed in the presence of ligand (Fig 1C and S3 Fig).

Treatment with progesterone resulted in AR mRNA reduction in the presence of either PR-A ($p = 0.038$) or PR-B ($p = 0.025$) (Fig 1D). Contrary to ERβ, the effect of progesterone on the AR transcript was dependent upon both the presence of hormone and the receptor protein (S4 Fig). Importantly, the primers used are specific to the AR transcript, as no other human transcripts, including those of the other steroid receptors, align with sufficient specificity for

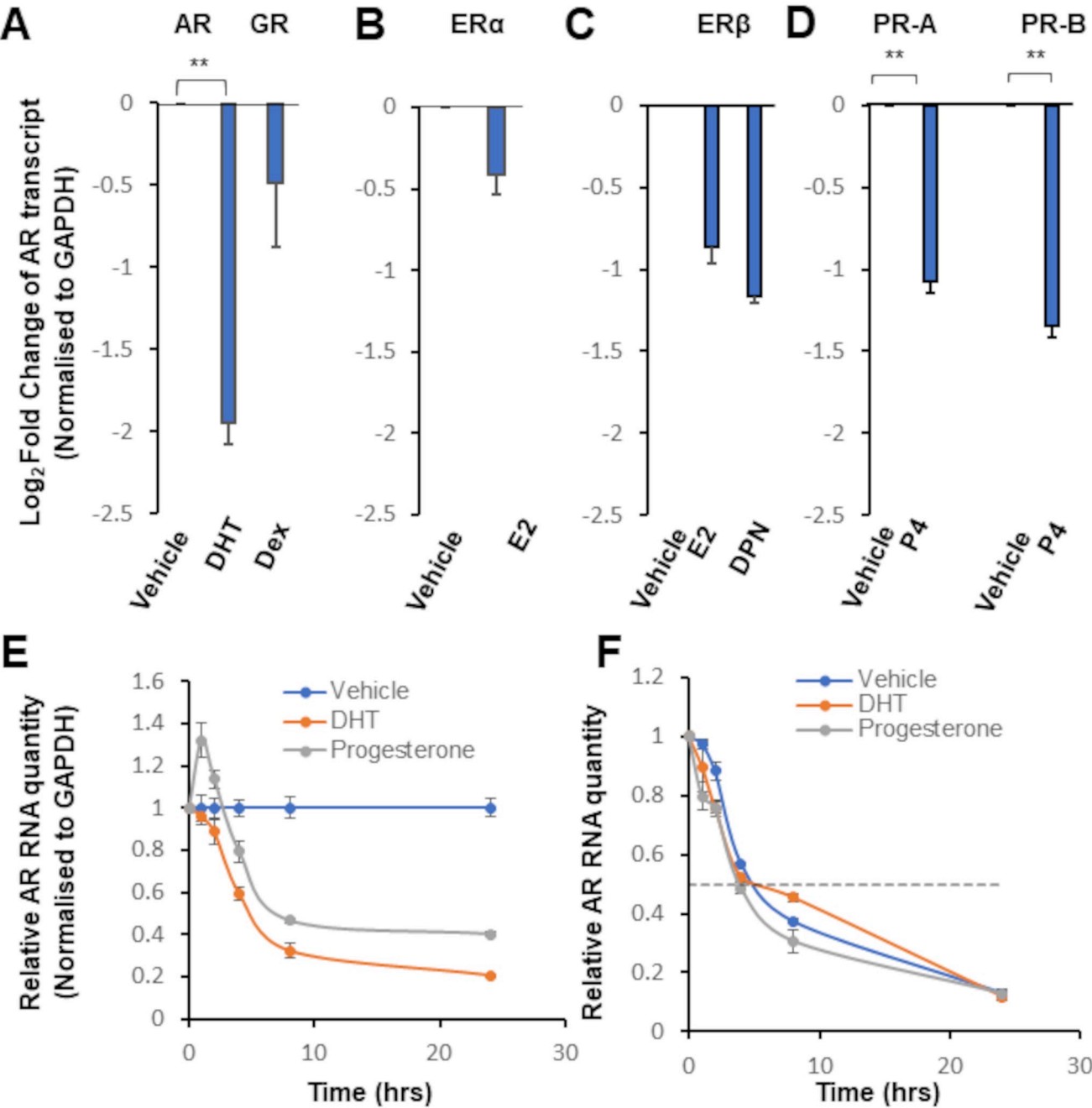

**Fig 1. Activated steroid hormone receptors repress androgen receptor expression.** Androgen receptor transcript levels in VCaP cells, as measured by qRT-PCR. (**A–D**) Cells were (**A**) untransfected, (**B**) transfected with pCDNA3-ERα, (**C**) transfected with pDC315-ERβ-CFP or (**D**) transfected with CMV-hPR-A and CMV-hPR-B as shown. Cells were treated for 24 hours with 10 nM hormone or vehicle (Ethanol). GAPDH was used as an endogenous control. (**E**) Cells were treated with vehicle or 10 nM hormone at t = 0. Cells treated with progesterone were transfected with CMV-hPR-B. (**F**) Cells were pre-treated with vehicle or 10 nM hormone for 2 hours prior to addition of 10 μM actinomycin. Cells treated with progesterone were transfected with CMV-hPR-B. Dotted line shows half maximal RNA. DHT–dihydrotestosterone. Dex–dexamethasone. E2–17β-oestradiol. DPN–diarylpropionitrile. Pro–progesterone. In all cases, error bars show standard error of the mean (3 biological replicates). ** = p < 0.01 (Unpaired Student's t-test).

PCR amplification; and possible genomic targets are not only poorly aligned to the primers, but are also far too large to account for the product size noted (S5 Fig).

To better understand the role of androgen and progesterone in AR regulation, we investigated how transcript levels varied across a 24-hour time period following activation of AR or PR (Fig 1E). Both hormones reached maximal inhibition between 8- and 24-hour treatment, although DHT reduced AR transcripts to a lower level. Intriguingly, the time-course of repression for each receptor was different. AR activation led to a decrease in transcript levels at all stages as compared to equivalent vehicle-treated samples, while progesterone treatment increased AR expression at the 1-hour time point, before decreasing transcript levels back to and then below baseline levels. A difference in RNA levels does not intrinsically imply altered gene expression but may instead indicate altered RNA stability. To distinguish between these possibilities, we treated cells with the transcription inhibitor actinomycin D, in the presence of each hormone. As transcription was blocked, variation in RNA levels was dependent on RNA stability (Fig 1F). We found that AR transcript half-life was not significantly different following treatment with either vehicle (4.4 ± 0.5 hrs), DHT (4.1 ± 0.4 hrs), or progesterone (3.5 ± 0.4 hrs), indicating hormone-treatment did not appreciably affect the stability of the AR transcript.

Splice variants, arising from cryptic exons, lacking the ligand binding domain of the AR, have been reported in several cell models, notably the prostate cell lines VCaP and 22rv1 [40]. We therefore tested the effect of hormone treatment on splice variant mRNA levels. Treatment of 22rv1 and VCaP cells with DHT led to a significant decrease in transcript levels for ARv1 (S6A Fig) and ARv7 (S6B Fig). Furthermore, treatment with progesterone also resulted in a significant decrease in ARv7 in VCaP cells (S6B Fig). Taken together our results demonstrate a hormone-dependent repression of AR full-length and variant mRNAs in prostate cells.

To investigate further the effects of hormone on AR-mediated signalling, protein expression of AR protein and the AR-target PSA were analysed by Western blotting (Fig 2). Treatment of VCaP cells with DHT significantly (p = 0.011) reduced AR protein levels, as did progesterone treatment in the presence of both PR isoforms (p = 0.019) although not in the presence of PR-A alone (Fig 2B). It is worth noting that in several cell models, notably in LNCaP, androgen treatment leads to increased stability and levels of AR protein. However, in VCaP it has been observed that hormone-treatment leads to a decrease as seen here [see also 15]. In terms of downstream AR targets, DHT significantly increased PSA levels (p = 0.02) as expected. Progesterone, by contrast, significantly (p = 0.043) decreased PSA protein in cells transfected with CMV-hPR-B (Fig 2C).

## 3.2 Both AR and PR bind to known AR control elements upon hormone activation

The AR has previously been shown to bind to two separate regions [15, 17] within the *AR* gene, but the relative contributions of these sites to the downregulation of the AR mRNA remains unclear. Given the homology between PR and AR, particularly in terms of their capacity to bind similar response elements, and the reduction in AR mRNA in response to progesterone treatment, we hypothesised that progesterone-activated PR would bind to the same response elements within the AR gene as DHT-activated AR. To test this, chromatin immunoprecipitation was carried out using anti-AR and PR antibodies, followed by qPCR (ChIP-qPCR) to quantify recruitment of both receptors, in response to ligand, to the 5' UTR element and the intron 2 element.

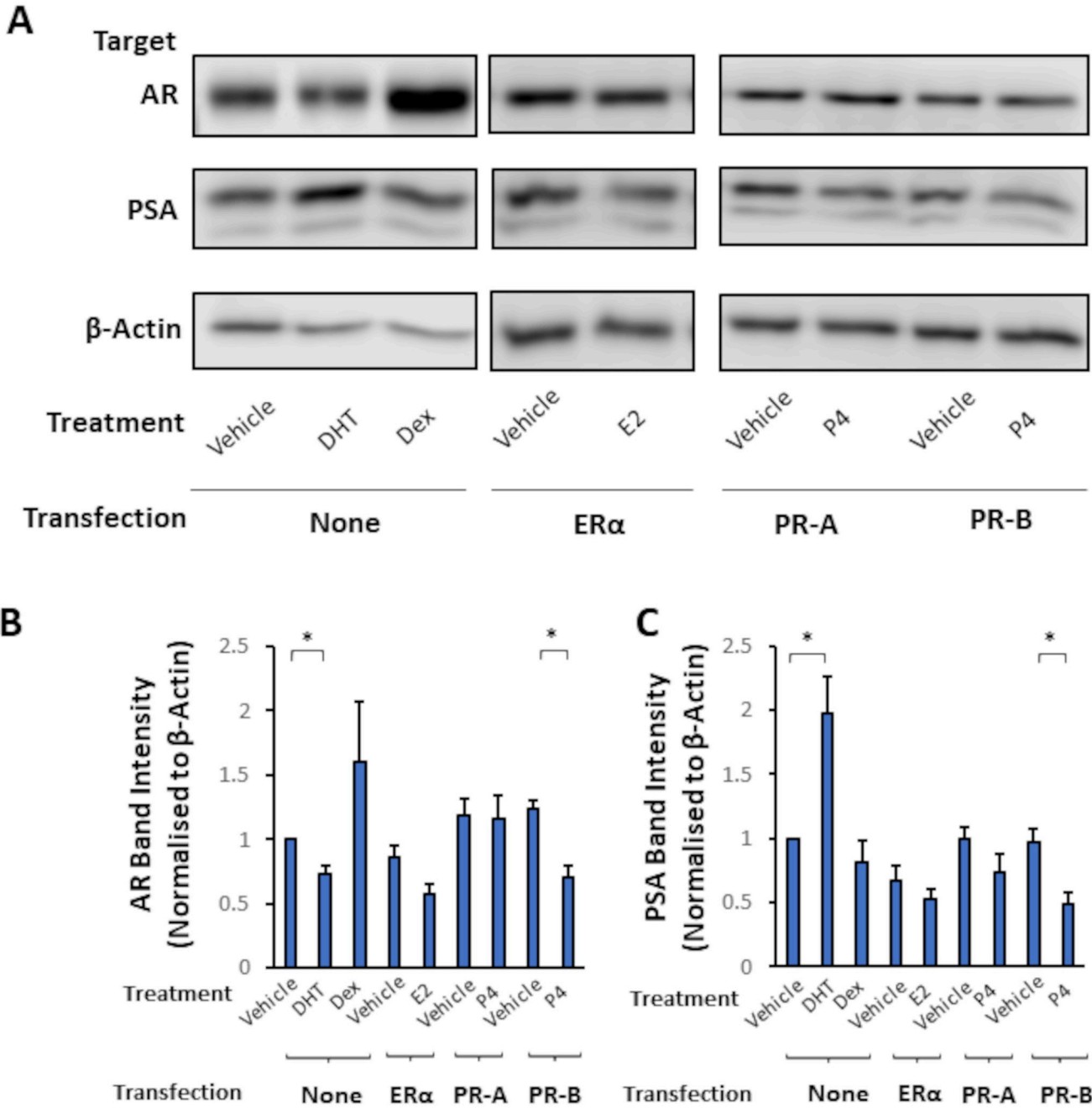

**Fig 2. Hormone signalling reduces AR protein and downstream targets.** (A) Representative images from Western Blots (n = 3) against given targets (3 biological replicates). VCaP cells were transfected with appropriate expression plasmids, then treated with 10 nM hormone or vehicle (Ethanol) for 24 hours. β-Actin shown as a loading control. (B and C) Quantification of protein levels in (B) AR and (C) PSA Western Blots normalised to β-Actin. Error bars show standard error of the mean. DHT = dihydrotestosterone. Dex = dexamethasone. E2 = 17β-oestradiol. P4 = Progesterone. * = p < 0.05 (Unpaired student's T-test, comparing protein normalised only to β-Actin).

We noted significant recruitment of AR to both the 5' UTR (Fig 3A) and Intron 2 (Fig 3B) response elements following DHT treatment (p = 0.028 and p = 0.007 respectively). Interestingly, there was an approximate 6.5-fold increase in AR recruitment to the intron 2 element, compared to a 2.5-fold increase in 5'UTR region. This suggests preferential recruitment of the

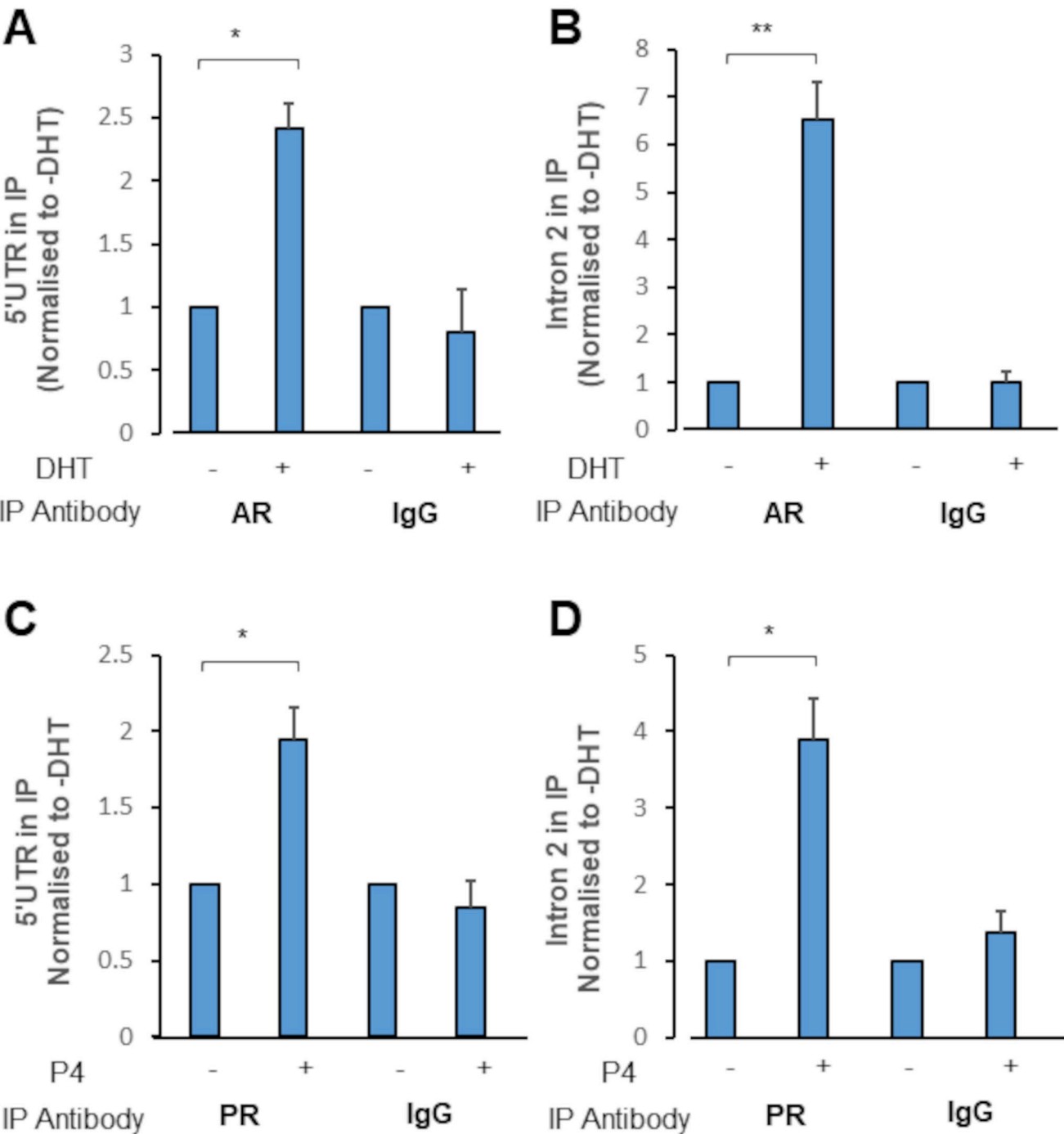

**Fig 3. Androgen and progesterone receptor are recruited to DNA sequences within the AR gene.** (**A** and **B**) VCaP cells were treated with vehicle (Ethanol) or 10 nM DHT for 4 hours. Chromatin was harvested, fragmented by sonication, and immunoprecipitated with the appropriate antibody (IgG used as a non-binding isotype control). Precipitated DNA quantified via qPCR using primers for (**A**) the 5' UTR Element or (**B**) the intron 2 Element. (**C** and **D**) As above, but VCaP cells were first transfected with CMV-hPR-B 24 hours prior to treatment with 10 nM progesterone or vehicle (ethanol). qPCR results shown for (**C**) the 5' UTR element and (**D**) the Intron 2 Element. In all cases, error bars show standard error of the mean (3 biological replicates). * = p < 0.05; ** = p< 0.01 (Unpaired Student's t-test).

AR to the intron 2 element following treatment with DHT, as primers to both the 5'UTR and the Intron 2 element amplified the DNA with comparable efficiencies (S7 Fig). ChIP for PR binding revealed a strikingly similar pattern of binding. We noted significant recruitment of the receptor to both the 5' UTR (Fig 3C) and intron 2 (Fig 3D) elements following hormone treatment, with recruitment to the latter element appearing 2-fold higher than the former (p = 0.018 and p = 0.012 respectively).

While both AR and PR bound to these sites within the AR locus, it does not necessarily follow that these response elements are active in AR downregulation. To infer functionality of the 5' UTR repressor, we utilised a luciferase construct, phAR1.6Luc [17], which contains 1.6 kb of the AR gene surrounding the transcriptional start site (TSS) upstream of a luciferase gene.

In LNCaP cells, DHT treatment caused a significant reduction in luciferase activity [17]. However, LNCaP cells were deemed inappropriate, for the comparison with PR, as they possess an AR ligand binding domain mutation (T878A) which has promiscuous progesterone and oestradiol-mediated activation [41, 42]. Therefore, PC3 cells were transfected with wild-type AR or both PR isoforms and the phAR1.6luc reporter gene. DHT treatment resulted in a significant (p = 0.007) reduction in luciferase activity, as expected (Fig 4A).

We also observed a significant reduction in luciferase activity following progesterone treatment regardless of which PR expression vector was present in the cells (p = 0.018 for PR-A; p = 0.02 for PR-B), although the magnitude of reduction is lower than noted with DHT (Fig 4B). These results support the conclusion that binding of either AR or PR to the 5' UTR element results in a downregulation of AR expression in a hormone-dependant manner.

To investigate further the potential regulatory feedback loops involving the PR and the AR gene, we probed a limited set of tissues that had data sets for both AR and PR binding. In the case of the PR, we found that ChIP-seq for either PR isoform in endometrial stroma resulted in increased read depth within the putative 5' UTR element compared to no immunoprecipitation controls, although this did not reach statistical significance (S8A and S8B Fig). This lack of statistical support may be related to the fact that only a single replicate exists for each condition in the analysed dataset [32]. In patient leiomyoma samples [33], we found a peak upstream of the AR gene which was significantly enriched in anti-PR ChIP compared to no immunoprecipitation controls (S8A and S8C Fig). This peak was in an almost identical position to that detected in AR ChIP in breast cancer cells [31] (S10A Fig). However, no PR binding to the AR 5' UTR was noted in response to synthetic progestogen treatment in breast cancer cells (S8A and S8D Fig).

Investigation of both PR-A and PR-B ChIP-seq datasets demonstrated a qualitative increase in read depth within the AR intron 2 regulatory element compared to non-immunoprecipitated samples of normal endometrial stroma, though again this did not reach statistical significance (S9A and S9B Fig). However, there was no evidence of PR binding at the intron 2 element in leiomyoma patient samples (S9A and S9C Fig) while in breast cancer cell lines, there is strong evidence for PR binding to the Intron 2 regulatory element (S9A and S9D Fig). Comparing vehicle-treated samples with those treated with the synthetic progestin R5020, there is a marked increase in read depth at the Intron 2 regulatory element. This is supported statistically by MACS2 analysis, which demarcates almost the whole region of interest as a significantly enriched peak. Interestingly there was little evidence of AR binding the intronic element in either endometrial stroma or breast cancer cells [30, 31] (S10B Fig). Collectively, these results and those described in the present study support a mechanistic model for PR regulation of the AR gene in different cell types and/ or tissues and the importance of sequences surrounding the TSS, 5'UTR and intron 2.

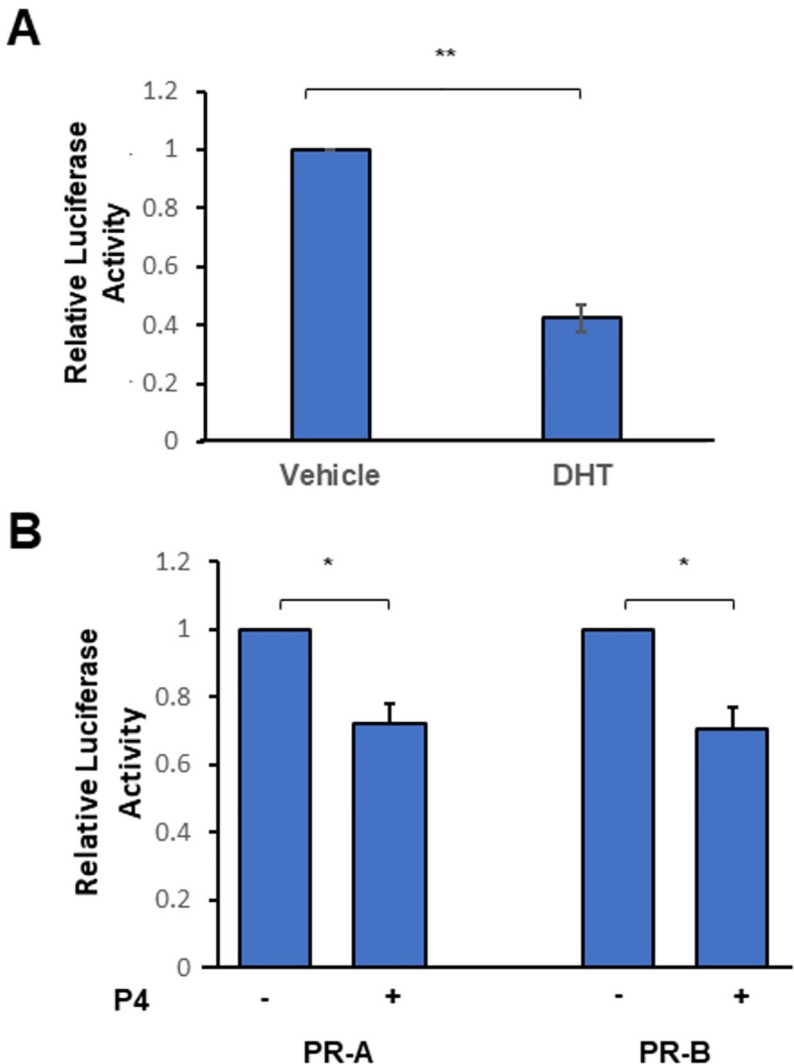

**Fig 4. AR and PR binding to the 5'UTR element drives AR trans-repression.** (**A**) PC3 cells were transfected with the pARo and phAR1.6Luc reporter plasmid, then treated with vehicle or 1 nM DHT for 24 hours. (**B**) PC3 cells were transfected with phAR1.6Luc reporter plasmid plus either CMV-hPR-A or CMV-hPR-B. These cells were then treated with vehicle or 10 nM P4 for 24 hours. Luciferase activity is shown relative to vehicle treated samples, normalised for total protein. Error bars show standard error of the mean (3 biological replicates). * = p < 0.05; ** = p< 0.01 (Unpaired Student's t-test).

### 3.3 Hormonal alteration of the AR epigenetic landscape

We next investigated the mechanisms underpinning the observed down regulation of the AR transcript by AR and PR. Given previous findings that histone modifications may play a role in androgen-mediated control of *AR* expression [15, 16], we tested the likelihood that the receptors recruit histone modifying enzymes as effectors. ChIP-qPCR was carried out, utilising antibodies against modifications on histone 3, including di- and tri- methylation of lysine residue 4 (H3K4me2 and H3K4me3 respectively) and acetylation of lysine residues across the protein (H3Ac), specifically at residues K9, K14, K18, K23 and K27, which are known to be a mark of activate transcription. Immunoprecipitation of chromatin samples from hormone treated VCaP cells with these antibodies, followed by qPCR, allowed us to monitor changes in

histone mark deposition at both negative regulatory regions following hormone treatment (Fig 5).

Treatment with DHT significantly (p = 0.003) decreased H3Ac marks in the 5' UTR (Fig 5A), whilst having no effect on H3K4 methylation. By contrast, at the intron 2 element, H3K4me2 was significantly (p = 0.004) decreased following hormone treatment, while H3Ac was unaffected. Thus, supporting a role for different histone modifying enzymes at these regulatory regions. In both cases, H3K4me3, which is modified by different enzymes to H3K4me2, was unaffected by DHT treatment.

Progesterone-mediated changes in histone marks were measured using VCaP cells transfected with CMV-hPR-B. Crucially, this transfection did not in itself significantly affect any of the histone marks studied (S11 Fig). Similar to androgen-mediated changes, treatment of these cells with progesterone resulted in reduced H3Ac within the 5' UTR (Fig 5B, p = 0.014). In contrast, no significant differences in any of the studied histone marks was observed at the intron 2 element following progesterone treatment. Taken together, these data support mechanistic differences between androgen- and progesterone-mediated repression of AR mRNA expression.

To further determine whether histone deacetylase (HDAC) activity was necessary for hormone-mediated AR repression we utilised two different inhibitors, SAHA (a Class I/ClassII HDAC inhibitor) and Sodium Butyrate (a Class I specific HDAC inhibitor). VCaP cells were treated with vehicle, inhibitor, along with either ethanol, DHT or progesterone for 24 hours and AR expression was measured via qRT-PCR (Fig 6). SAHA treatment did not significantly affect repression of the AR transcript in the presence of either DHT or progesterone. In contrast, a significant increase in AR transcript was noted in the presence of sodium butyrate following activation of either the AR (p = 0.022) or PR (p = 0.017). Indeed, in the latter case, the presence of Sodium Butyrate fully restores AR transcript to baseline levels. The distinct effects of these inhibitors indicate that specific inhibition of Class I HDACs, but not Class II HDACs, results in a rescue of AR mRNA expression. Importantly, the concentrations of SAHA (50 nM) and sodium butyrate (10 μM) did not impact on basal levels of AR mRNA expression (S12A and S12B Fig). These results strongly support a role for Class I HDAC activity in the hormone-mediated repression effected by both PR and AR binding within the 5'UTR of the *AR* gene.

## 3.4 Long range chromatin interactions

Given that the intron 2 site lies over 100 kb downstream of the AR transcriptional start site (TSS), long-range interactions must be facilitated by hormone receptor binding to this element to repressor activity. To investigate this possibility, we designed a ChIP-Loop experiment, using anti-AR or anti-PR antibodies to isolate receptor bound DNA, followed by on-bead ligation to connect DNA fragments. An arbitrary upstream control region was designated at approximately 80 kb upstream of the TSS, to allow comparison with a region which should not loop, regardless of hormone activation. To detect any possible looping, PCR was carried out with one primer situated at the TSS, and another in either the intron 2 element or the upstream control region (Fig 7A). A control ligation library was generated containing all possible ligation products across our region of interest using two artificial chromosomes–the PAC RP1-22L8 and the BAC RP11-479J1. Control ligations mix PCR with either TSS—Intron 2 or TSS—Upstream Control primers showed effective amplification in each case, with no amplification in internal negative controls.

ChIP-Loop using anti-AR pulldown for the TSS—intron 2 chimeric DNA molecule identified a product only in the presence of DHT, indicating hormone-dependent chromatin

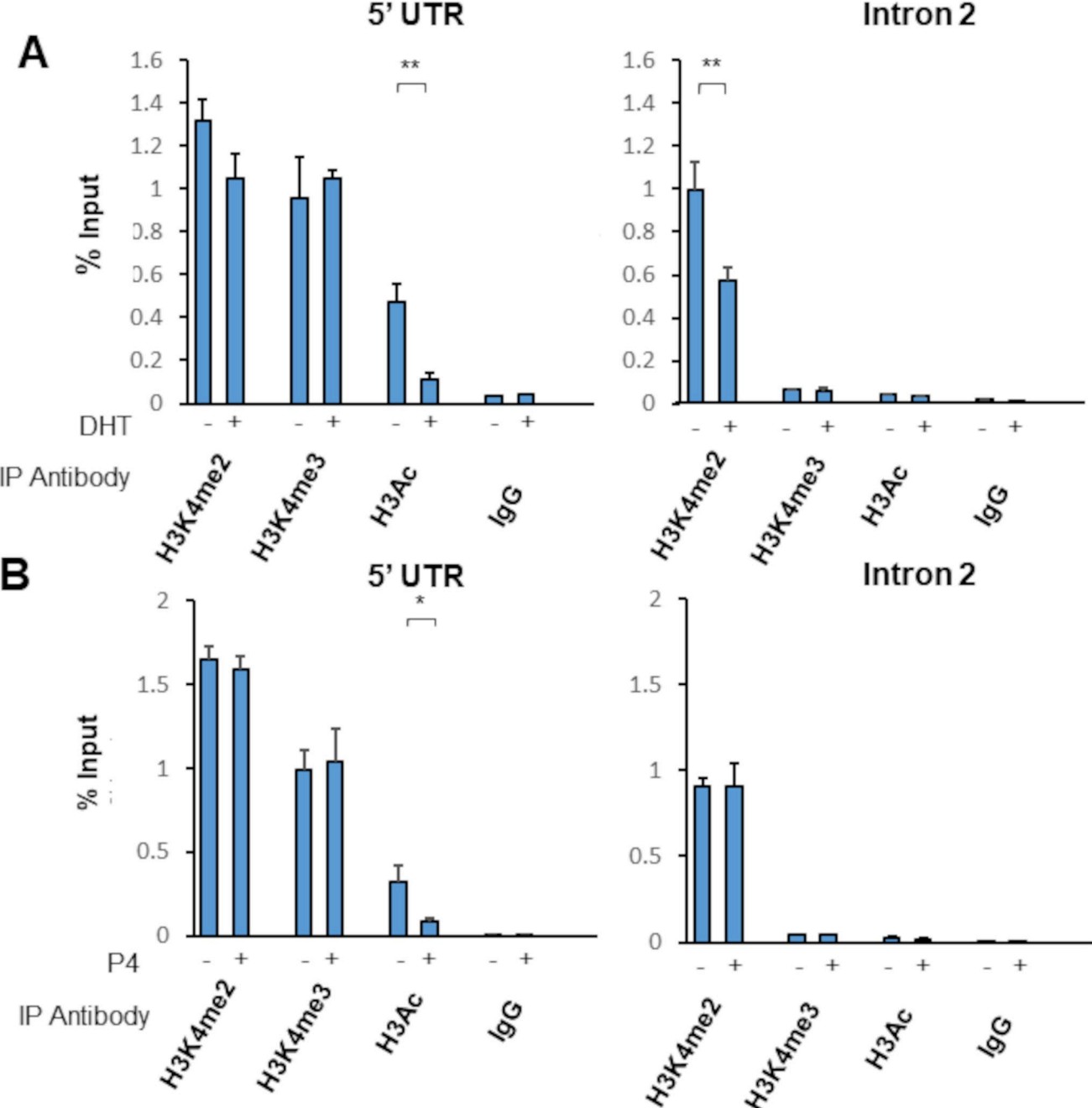

**Fig 5. Hormone treatment affects histone marks at both 5'UTR and Intron 2 regulatory regions.** (**A**) VCaP cells were treated with vehicle (ethanol) or 10 nM DHT for 4 hours. Chromatin was harvested, fragmented by sonication, and immunoprecipitated with the appropriate antibody (IgG used as a non-binding isotype control). Precipitated DNA quantified via qPCR using primers for the 5' UTR element (left panel) or the intron 2 element (right panel). (**B**) As above, but VCaP cells were first transfected with CMV-hPR-B for 24 hours, then treated with vehicle (ethanol) or 10 nM progesterone for 4 hours. qPCR used primers for (**C**) the 5' UTR element or (**D**) the Intron 2 element. Error bars show standard error of the mean (3 biological replicates). * = p<0.05; ** = p< 0.01 (Unpaired Student's t-test).

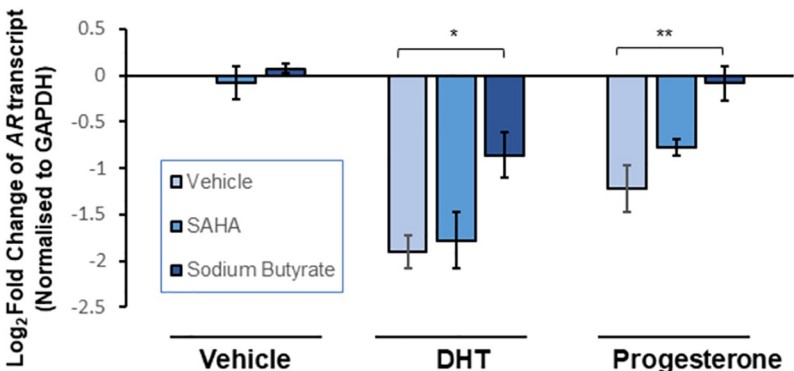

**Fig 6. Histone deacetylation facilitates repression of the AR gene.** Androgen receptor transcript levels in VCaP cells, as measured by qRT-PCR. Cells were treated with 10 nM of appropriate hormone or vehicle (ethanol), plus transfection of pCMV-hPR-B in progesterone treated cells. At the same time, cells were treated with either vehicle ($H_2O$), 50 nM SAHA or 10 µM sodium butyrate. RNA was harvested 24 hours after treatment. In all cases, GAPDH used as an endogenous control. Error bars show standard error of the mean (3 biological replicates). * = $p < 0.05$. ** = $p < 0.01$ (ANOVA with Tukey's post-hoc test).

looping (Fig 7B). Equivalent experiments investigated the effect of progesterone (P4) treatment, following transfection hPR-B. Chromatin immunoprecipitation with anti-PR antibody resulted in a hormone-dependent loop, joining the intron 2 repressor region and the AR TSS (Fig 7C). Both DHT- and progesterone- mediated looping was abolished in ligase negative samples, confirming that this product was not an artefact generated by inefficient digestion as there was an absolute requirement for ligation of separate restriction fragments. Looping to the designated upstream control sequence was not detected in any sample, verifying that hormone-mediated looping was specific to the intron 2 response element (Fig 7B and 7C). Thus, binding of the ligand-activated AR or PR promoted looping between the 5'UTR and intron 2 leading to the physical proximity to the TSS of the *AR* gene.

## 4. Discussion

The Androgen receptor is a hormone activated transcription factor whose activity is important in both male and female reproductive and non-reproductive tissues and cells [5–8, 43]. AR signalling depends largely on *AR* transcript levels, with AR overexpression leading to alteration of downstream transcriptional control [44, 45]. Thus, understanding androgenic signalling requires an in-depth analysis of the molecular mechanisms controlling *AR* expression. This is particularly relevant in the context of hormone-dependent cancers, and disorders such as polycystic ovarian syndrome (PCOS) and endometriosis (see [6, 46] and references therein). In this study, we sought to clarify the pathways underpinning hormonal control of androgen receptor expression. Crucially, we explored not only the well-documented phenomenon of AR autoregulation, but also identified a hitherto unreported repression mechanism via the progesterone receptor in a prostate epithelial cell model.

ChIP analysis confirmed identical genomic targets for both AR and PR within the AR gene–one in the 5'UTR and one in intron 2. Intriguingly, our data suggests that both receptors are recruited preferentially to the Intron 2 element upon hormone treatment. Previous experiments have proven recruitment of the AR to both elements, but never in a manner that allowed direct comparison [15–17]. This suggests that despite an absence of canonical binding sites, the intron 2 response element could be the preferential binding site for hormone receptors.

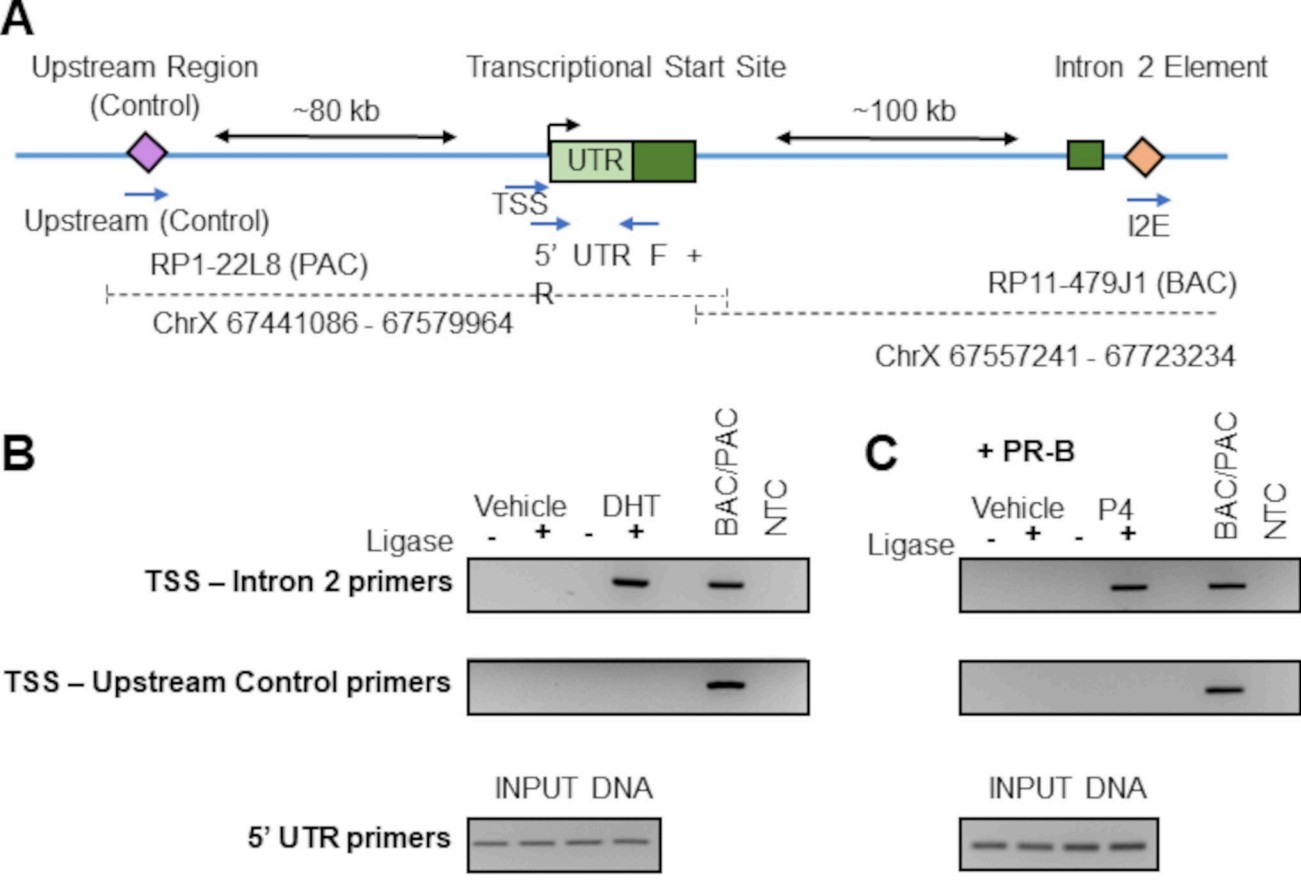

**Fig 7. Chromosomal looping of the promoter (TSS) and intron 2 regions is steroid hormone dependent.** (**A**) The genomic context surrounding the AR transcriptional start site (TSS, located at ChrX 66763863). The intron 2 element and ChIP-Loop upstream control regions are indicated, along with approximate distances from the TSS. Below, ChIP-Loop primer positions are noted, along with traditional PCR primers for detection of the 5' UTR element, which will not be disrupted by BstY1 digestion. The relative positions of two artificial chromosomes used to generate ChIP-Loop control libraries are also shown. The 5'UTR is shown as a pale green box, whilst exons are dark green. (**B**) VCaP samples were treated with 10 nM DHT for 4 hours, then chromatin was fixed, harvested, and looping analysed via anti-AR ChIP-Loop. Samples used as template for PCR reactions with the indicated primers, alongside BAC/PAC control ligation mix as a positive control and a no-template control (NTC). Input DNA (pre-immunoprecipitation) was run in PCR for the 5' UTR element, which is not disrupted by BstY1 digestion. PCR products were run on 2% Agarose gels. (**C**) As for **B**, except VCaP cells were transfected with CMV-hPR-B, then treated with 10 nM progesterone and immunoprecipitation carried out with anti-PR antibody. Images are representative of 3 biological replicates.

Luciferase reporter gene assays confirmed that binding to the 5' UTR element reduced transcriptional activity, although the poorly defined nature of the intron 2 element precluded similar analysis. However, the finding that receptor-dependent looping brings intron 2 into close proximity to the *AR* gene TSS/5'UTR, upon stimulation with either androgens or progestogens, provides a mechanism for repression, integrating both negative regulatory regions (Fig 8).

Analysis of histone mark deposition at hormone response elements identified H3K4 demethylation at the intron 2 element upon AR activation, as has been previously reported [15, 16]. This mark was associated with recruitment of Lysine Demethylase 1 (LSD1 or KDM1A) to the AR gene, which removes activating H3K4 mono- and di-methyl marks [47] and identifies one mechanism by which AR may downregulate its own expression. However, we also note a significant downregulation of histone 3 acetylation marks at the 5'UTR element

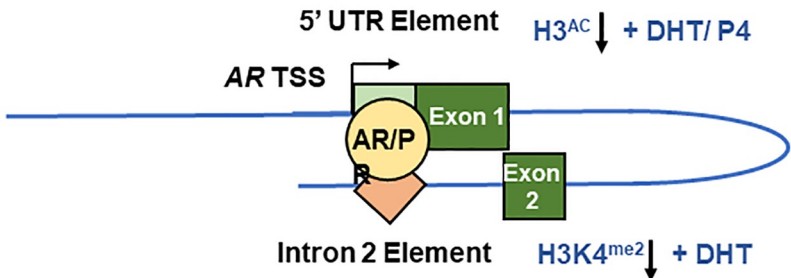

**Fig 8. Model for steroid hormone-mediated AR gene repression.** AR and PR are each recruited to the 5'UTR and intron 2 elements, which then loops back to allow physical interaction with the AR TSS. This results in recruitment of histone modifying enzymes. Both receptors recruit HDACs to reduce H3ac at the 5' UTR, whilst only AR reduces H3K4me2 at the intron 2 region.

while H3K4 methylation remains unchanged suggesting an LSD1-independant mechanism for AR repression via the 5' UTR. This supports previous findings from AR ChIP-LSD1 reChIP analysis of the AR 5' UTR, which found no evidence of AR-driven LSD1 recruitment to this site [17].

Acetylation of lysine residues within histone tails neutralizes their charge, reducing affinity for DNA to create an open chromatin state, facilitating gene expression [48]. Conversely, deacetylation reduces gene expression, catalysed by HDACs. HDAC activity is understood to be global rather than targeting specific residues, and thus the loss of H3Ac noted in this work upon hormone treatment implies a loss in acetylation of other histones [49, 50]. Histone deacetylation as a consequence of hormone signalling has been documented for both androgens [51, 52] and progestins [53]. Here, we demonstrate that HDAC activity is necessary to achieve the full inhibitory impact of hormone treatment. This is particularly true for progesterone-dependent AR repression, as inhibition of class I HDACs fully rescued AR expression in the presence of hormone.

Reduction of AR mRNA levels by DHT or progesterone, while appearing similar, demonstrate hormone-selective properties. In particular, AR more potently reduces AR transcript levels as compared to PR. Differential receptor recruitment to DNA may go some way to explaining this difference. Receptor recruitment to the 5' UTR element was somewhat similar, but recruitment to the intron 2 element was substantially lower for PR. This may be explained by the fact that the hormone receptor binding sites within the intronic region appear to be largely half-sites [16]. AR binds readily to half-site response elements, while PR capacity for half-site binding is more dependent on structural features of the DNA [54, 55]. Thus, the structure of the intron 2 repressor appears to favour the binding of AR over PR.

The effect of PR binding within intron 2 also differs significantly from the AR. While both receptors induce chromosomal looping in response to hormone, only the AR reduces H3K4me2 deposition, suggesting that progesterone-dependent repression of *AR* does not utilise LSD1. This may explain why AR and PR repress AR mRNA to differing degrees as the histone deacetylation activity is common to both receptors, but H3K4 demethylation is exclusive to AR and therefore the effect of AR on its own expression is more profound. Similarly, this goes some way to explaining the differing effects of HDAC inhibition on the receptors. If PR repression of AR is solely mediated by HDAC activity, it is logical that inhibiting this enzyme activity should result in full rescue of AR expression. On the other hand, androgenic autorepression appears to employ multiple pathways, allowing repression to remain at least partially active even when HDACs are inhibited.

In the normal endometrium of women and non-human primates, the androgen receptor is expressed in the stromal cells [56]. However, treatment with an anti-progesterone, RU486, results in a significant increase in the AR protein in both stromal and glandular epithelial cells [57]. Our analysis of publicly available data sets [30–34] identified recruitment of the PR to both 5'UTR and intron 2 sequences of the AR gene (S8 and S9 Figs). Thus, providing a mechanistic explanation for the impact of the anti-progesterone drug.

Recently, Hickey and co-workers [43] re-evaluated the role of the AR in breast cancer and provided compelling evidence for the use of an agonist to activate AR signalling as a therapeutic intervention in ER+ breast cancers. It is interesting therefore that the progesterone receptor has been shown to drive tumour growth and metastasis in the absence of ER activity in breast cancer [58]. While Garcia et al [59] reported that expression of the AR was more prevalent in ER+/PR- compared with ER+/PR+ tumours. Significantly, we found ligand-dependent recruitment of the PR in ChIP-seq data [34] for a breast cancer cell-line (S8 and S9 Figs). Thus, PR inhibition of the *AR* gene could further impact on the effectiveness of the AR as a therapeutic target and contribute to the tumour promoting activity of progestins.

The AR is a well described and validated drug target in advanced and metastatic prostate cancer although it may play distinct roles in both the stromal and epithelial compartments. Interestingly, there is renewed interest in the role of the AR in breast cancer subtypes as either a therapeutic target or prognostic biomarker [43, 59, 60]. Our results shed a mechanistic insight into the regulation of the *AR* gene in breast cancer cells, as absence of the PR is likely to abrogate the progesterone-mediated repression of the *AR* gene, while maintaining the positive regulation previously described for the ER in human endometrium [61] and mouse Fallopian tubes [62]. It will be important to test this model in cell lines expressing endogenous levels of both the AR and PR. Collectively, our data together with published studies [6, 32, 60] support distinct, but overlapping, regulatory networks for androgen and progesterone regulation of the *AR* gene in breast and prostate epithelial cells and endometrial stromal cells.

## Supporting information

**S1 Fig. Steroid hormone receptors in prostate cancer cell lines.** (A) Images of Western Blots for listed steroid hormone receptors in different prostate cell lines. β-Actin shown as a housekeeping gene and loading control. (B) Table summary of Western Blot data; +, expression, +/-, modest expression; and -, negative.
(PDF)

**S2 Fig. Transfection with oestrogen and progesterone receptors.** (A) Anti-ERα immunoblots of protein extracts taken from VCaP cells in the presence or absence of pCDNA3-ERα transfection. (B) Anti-PR immunoblots of protein extracts taken from VCaP cells transfected with CMV-hPR-A and CMV-hPR-B. The PR antibody used detects both PR isoforms. (C) Anti-CFP immunoblots of protein extracts taken from VCaP cells in the presence or absence of pDC315-ERβ-CFP. β-Actin shown as a loading control in all cases. Note a band occurs in the untransfected cells with the αCFP antibody, suggesting cross-reactivity with cellular proteins.
(PDF)

**S3 Fig. Transfection with estrogen receptor β affects the AR pathway in the absence of hormone.** (A) qRT-PCR of RNA samples harvested from VCaP cells treated with various ERβ ligands in the presence or absence of pDC315-ERβ-CFP. GAPDH used as an endogenous control for data normalisation. Error bars show ± standard error of the mean (3 biological replicates). ** = $p < 0.01$ (ANOVA with Tukey's Post-Hoc test). (B) Anti-AR and Anti-PSA

immunoblots of protein extracts taken from VCaP cells treated with various ERβ ligands in the presence or absence of pDC315-ERβ-CFP. β-Actin shown as a loading control. Blots are representative of 3 biological replicates. E2 = 17β-oestradiol. DPN = diarylpropionitrile.
(PDF)

**S4 Fig. Progesterone only affects *AR* transcription via the activated progesterone receptor.** qRT-PCR of RNA samples harvested from VCaP cells treated with vehicle (Ethanol) or 10 nM progesterone (P4) for 24 hours, in the absence (-PR) or presence of hPRB (+ PR); GAPDH used as an endogenous control for data normalisation. Error bars show ± standard error of the mean (3 biological replicates). * = p < 0.05 (Unpaired Student's t-test).
(PDF)

**S5 Fig. Androgen receptor primers are specific to the AR transcript.** (A) PCR products from AR amplification were run on a 2% agarose gel along with a non-target control (NTC) sample and a 100 bp DNA Ladder (New England Biolabs). Ladder band sizes are marked. (B and C) NCBI Primer BLAST results against all Refseq *Homo sapiens* mRNA and Refseq *Homo sapiens* genome sequences respectively. These tables show every sequence to which primers align with <5 mismatches per primer.
(PDF)

**S6 Fig. Down-regulation of full-length and receptor splice variant mRNA.** (A) In 22Rv1 cells, transcripts containing cryptic exons (CEs) 1, 3 and 5 were all detected, albeit to differing levels. To interrogate the effects of androgen signalling on AR-v levels we compared transcript levels in cells treated with either vehicle or 10 nM hormone for 24 hours by qRT-PCR. (B) In VCaP cells, it was possible to treat with both DHT and, in the presence of PR-B, progesterone. Consistent with earlier experiments, we noted repression of AR-FL transcripts in the presence of either hormone (A and B). The same was true of AR-V1 (22rv1 cells) and AR-V7 (VCaP cells) transcript levels when compared to vehicle control. Treatment with either DHT (p = 0.009) or progesterone (p = 0.017) resulted in a significant reduction in expression of CE3-containing transcripts.
(PDF)

**S7 Fig. Primer efficiencies for putative AR repressor sequences.** Example qPCR 5-fold dilution series data for primers against (**A**) the AR 5' UTR and (**B**) the AR Intron 2 element. Dilution series were carried out using Input ChIP samples. Ct value (average of 2 technical replicates) is plotted against dilution level across the dilution series. In both cases, this dynamic range covers experimental Ct values. qPCR efficiencies are shown as calculated by the formula ***efficiency* = −1 + 10**$^{(-1/slope)}$ to the nearest integer. These efficiencies fall within the 90%-110% acceptable range as outlined within the MIQE guidelines (*Bustin et al. The MIQE guidelines*: *minimum information for publication of quantitative real-time PCR experiments. Clin. Chem. 2009; 55, 611–622*).
(PDF)

**S8 Fig. Hormone receptor binding to the AR 5'UTR in other tissue types.** Publicly available ChIP-seq datasets were used to interrogate binding of progesterone receptor to the *AR* 5'UTR. A 3 kb window is shown, surrounding the putative 5' UTR receptor binding element (5UE), which is located between the dotted lines. Coverage of this region is shown for each experiment, with peak height corresponding to read depth (see scale on left of each bar). In all cases, peaks which differ significantly between conditions, as identified by MACS2, are indicated by blue box. (A) shows the genomic region of interest (hg19 chrX:66762600–66765599) to provide scale for ChIP-seq interpretation and the 5'UTR (5UE) region (chrX:66764465–

66764686) is highlighted. Below this, PR data sets are explored. These include, (B) normal endometrial stroma, with ChIP-seq data from either PR-A (SRR1614984); PR-B (SRR1614985) or no IP (SRR1614984) datasets. (C) Leiomyoma (endometrial tumour) patient samples, from either anti-PR (SRR10189640) or no IP (SRR10189645) datasets. (D) Breast cancer cell line samples, with ChIP-seq against PR in the presence of 1 nM R5020 (SRR15064561) or DMSO vehicle (SRR15064553).
(PDF)

**S9 Fig. Hormone receptor binding to the AR Intron 2 in other tissue types.** Publicly available ChIP-seq datasets were used to interrogate binding of the progesterone receptor to the *AR* Intron 2. A 3 kb window is shown, surrounding the putative Intron 2 receptor binding element (I2E), which is located between the dotted lines. Coverage of this region is shown for each experiment, with peak height corresponding to read depth (see scale on left of each bar). In all cases, peaks which differ significantly between conditions, as identified by MACS2, are indicated by blue boxes, with the dotted lines indicating the 12E region of interest. (A) Shows the genomic region of interest (hg19 chrX:66865500–66868499) to provide scale for ChIP-seq interpretation. The intron 2 region (I2E) (chrX:66866941–66867339) is highlighted. Below this, PR experiments are explored. These include, (B) normal endometrial stroma, with ChIP-seq data from either PR-A (SRR1614984); PR-B (SRR1614985) or no IP (SRR1614984) datasets. (C) Leiomyoma (endometrial tumour) patient samples, from either anti-PR (SRR10189640) or no IP (SRR10189645) datasets. (D) Breast cancer cell line samples, with ChIP-seq against PR in the presence of 1 nM R5020 (SRR15064561) or DMSO vehicle (SRR15064553).
(PDF)

**S10 Fig. Androgen receptor binding to the AR 5'UTR and Intron 2 regions in other tissue types.** (A) Publicly available ChIP-seq datasets were used to interrogate AR binding to the 5'UTR (A) and intron 2 sequences of the *AR* gene (B). Peaks which differ significantly between conditions, as identified by MACS2, are indicated by solid blue box. For the AR we analysed PCOS Patient endometrial stroma (SRR7782796), AR ChIP-seq only, and Breast Cancer cell line samples. AR ChIP-seq data is shown from cells treated with 10 nM DHT (SRR12626838) and DMSO vehicle (SRR12626837). A no-IP input sample is also shown for comparison (SRR12626843). We observed little evidence of AR binding at the 5' UTR element in endometrial stroma (Part A, Top Line). However, in data derived from breast cancer cells, we found a peak which was significantly enriched in anti-AR ChIP compared to no immunoprecipitation controls, suggesting that AR binds proximal to the transcriptional start site of the *AR* gene. (Part A, Lines 2 to 4). This peak did not overlap with the putative 5' UTR element we found enriched in prostate cells in response to hormone treatment. No evidence of AR binding to the I2E region was observed in either endometrial stromal samples or breast cancer cells (Part B).
(PDF)

**S11 Fig. Expression of progesterone receptor B does not affect histone marks at the *AR* gene.** VCaP cells were either transfected with CMV-hPR-B or remained untransfected. 24 hours later, chromatin was harvested, fragmented by sonication, and immunoprecipitated with the appropriate antibody (IgG used as a non-binding isotype control). Precipitated DNA quantified via qPCR using primers for (A) the 5' UTR Element or (B) the intron 2 Element. Error bars show standard error of the mean (3 biological replicates).
(PDF)

**S12 Fig. Titration of HDAC inhibitors.** VCaP cells were treated with increasing concentrations of either SAHA (A) or sodium butyrate (B). RNA was harvested 24 hours after treatment.

In all cases, GAPDH used as an endogenous control.
(PDF)

**S13 Fig. Uncropped blots for data presented in Fig 2.**
(PDF)

**S14 Fig. Uncropped blots for data presented in S1 Fig.**
(PDF)

**S15 Fig. Uncropped blots for data presented in S2 and S3 Figs.**
(PDF)

**S16 Fig. Uncropped gel images for data presented in S7 Fig.**
(PDF)

**S1 Table. A list of antibodies used in this study.**
(PDF)

**S2 Table. List of primers used in qPCR reactions.**
(PDF)

## Acknowledgments

We wish to thank the Aberdeen qPCR core facility for invaluable assistance in experimental design and execution. We would also like to thank Professors P. Saunders (University of Edinburgh), K.Dahlman-Wright (Karolinska Institute) and D.Edwards (Baylor College of Medicine) for providing expression plasmids for ERα, ERβ and PR-A/B respectively.

For the purpose of open access, the author has applied a Creative Commons Attribution (CC BY) [or other appropriate open licence] licence to any Author Accepted Manuscript version arising from this submission.

## Author Contributions

**Conceptualization:** Andrew D. Gillen, Iain J. McEwan.

**Data curation:** Irene Hunter, Iain J. McEwan.

**Formal analysis:** Andrew D. Gillen, Irene Hunter, Iain J. McEwan.

**Funding acquisition:** Iain J. McEwan.

**Project administration:** Iain J. McEwan.

**Supervision:** Ekkehard Ullner, Iain J. McEwan.

**Writing – original draft:** Andrew D. Gillen, Iain J. McEwan.

**Writing – review & editing:** Andrew D. Gillen, Irene Hunter, Ekkehard Ullner, Iain J. McEwan.

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
