## [Decision Letter · Decision Letter 0]

18 Sep 2023

PONE-D-23-27820Mechanistic Insights into Steroid Hormone-mediated Regulation of the Androgen Receptor GenePLOS ONE

Dear Dr. McEwan,

Thank you for submitting your manuscript to PLOS ONE. After careful consideration, we feel that it has merit but does not fully meet PLOS ONE’s publication criteria as it currently stands. Therefore, we invite you to submit a revised version of the manuscript that addresses the points raised during the review process.

We look forward to receiving your revised manuscript.

Kind regards,

Lucia R. Languino, Ph.D.

Academic Editor

PLOS ONE

Journal Requirements:

1. Please ensure that your manuscript meets PLOS ONE's style requirements, including those for file naming. The PLOS ONE style templates can be found at https://journals.plos.org/plosone/s/file?id=wjVg/PLOSOne_formatting_sample_main_body.pdf and https://journals.plos.org/plosone/s/fileid=ba62/PLOSOne_formatting_sample_title_authors_affiliations.pdf

2. Thank you for stating in your Funding Statement: "This work was supported by small project grants from NHS Grampian (17/040) and Friends of Anchor (RS 2019 001) to ADG and IJM. ADG was supported by a University of Aberdeen PhD studentship, warded competitively by the Institute of Medical Sciences. Funding for open access charge: None" 

5. Please note that in order to use the direct billing option the corresponding author must be affiliated with the chosen institute. Please either amend your manuscript to change the affiliation or corresponding author, or email us at plosone@plos.org with a request to remove this option.

6. We note that you have included the phrase “data not shown” in your manuscript. Unfortunately, this does not meet our data sharing requirements. PLOS does not permit references to inaccessible data. We require that authors provide all relevant data within the paper, Supporting Information files, or in an acceptable, public repository. 

Please add a citation to support this phrase or upload the data that corresponds with these findings to a stable repository (such as Figshare or Dryad) and provide and URLs, DOIs, or accession numbers that may be used to access these data. Or, if the data are not a core part of the research being presented in your study, we ask that you remove the phrase that refers to these data.

Reviewers' comments:

Reviewer's Responses to Questions

**Comments to the Author**

1. Is the manuscript technically sound, and do the data support the conclusions?

Reviewer #1: Partly

Reviewer #2: Partly

2. Has the statistical analysis been performed appropriately and rigorously? 

Reviewer #1: No

Reviewer #2: No

3. Have the authors made all data underlying the findings in their manuscript fully available?

Reviewer #1: No

Reviewer #2: Yes

4. Is the manuscript presented in an intelligible fashion and written in standard English?

Reviewer #1: Yes

Reviewer #2: Yes

5. Review Comments to the Author

Reviewer #1: The purpose of this study was to test which steroid receptors are able to negatively regulate AR gene transcription. Using a VCaP cell line model that either expresses endogenous steroid receptors (AR, GR, ERalpha) or was engineered to overexpress steroid receptors (PR-A, PR-B, ERbeta), the authors found that only AR, PR-A, and PR-B exerted this negative regulation. Negative AR autoregulation has been defined in the literature, but this represents the first report of PR-mediated negative regulation. ChIP-qPCR and promoter/repressor looping experiments indicated that AR and PR exert similar repression mechanisms by engaging with a hormone response element in AR intron 2 and promoting looping between this intron 2 repressor and the AR transcription start site. As elaborated below, there are a few concerns about the experimental design and data interpretation that may influence these conclusions. The authors are recommended to address these concerns to increase the accuracy of their study:

Major:

1. Because of the massive level of PR-A and/or PR-B overexpression in the VCaP model system, Figure 3 should include a ChIP-qPCR negative control region wherein PR-A and/or PR-B are not binding to DNA. It is possible that super-physiological overexpression of PR-A and/or PR-B could lead to a positive ChIP-qPCR signal no matter what region of the genome is targeted by qPCR.

2. The number of western blot replicates used for quantification and plotting in Figure 2 should be indicated in the Figure Legend. The additional western blot replicates used for densitometry should be included in the Supplement

3. The text on lines 343-349 is confusing and inconclusive. Why is the reduction in AR levels “striking” in Figure 1C, while not being significantly lower than ERbeta transfection alone in Supplementary Figure S3? Avoid the term “trend towards” on line 348. Overall, the authors should design their experiment and analyze their data in way that reveals a difference or does not reveal a difference (the null hypothesis).

4. A similar issue is noted for description of the public ChIP-seq data analysis in Fig S5 on lines 481-483 as well as lines 494-496. The results are not significant yet the authors use terminology like “increased read depth within the putative 5’ UTR element compared to no immunoprecipitation controls” and “qualitative increase in read depth within the AR intron 2 regulatory element compared to non-immunoprecipitated samples”. This type of data analysis lacks rigor.

5. A similar issue is noted on lines 557-558 “there was a trend towards increased AR levels in the latter case”

6. Figure 7B – the red boxes must be deleted as they obscure the gel area.

Minor:

7. The 3rd paragraph of the introduction (lines 86-102) should be updated to include discovery of the transcriptional enhancer ~650 kb upstream of the AR transcription start sites (PMIDs: 29909987, 30340047, and 30033370). Any interpretation of data demonstrating negative regulatory elements or suppressor/promoter looping would need to account for this knowledge.

8. The central model system used to generate the conclusions of this study is based on overexpression of PR-A or -B from a highly-active CMV promoter. Since the conclusions are all based on this overexpression model, this should be discussed as a limitation in the Abstract and in the Discussion.

9. Typo on line 375.

10. Past tense should be used throughout the Results section. There are numerous instances of switching to present tense.

11. Mistake on lines 478-479. The study is not evaluating the “PR and the AR gene”. The study is evaluating the effects of AR and PR activation on AR gene transcription.

Reviewer #2: This article addresses the role of the progesterone receptor as a regulator of AR expression. As a first approach, they used the prostate cancer cell line VCaP to individually test the effects of different steroid receptors on AR expression. Results showed that both AR transcript and protein are downregulated in progesterone-stimulated VCaP cells that ectopically express PR. Furthermore, they show that changes in transcriptional activity and not RNA stability are responsible for this outcome. Binding of PR was demonstrated at the 5’UTR region and the second intron of the AR gene, both known sites of AR binding and transcriptional repression. A gene reporter assay was done in the AR negative PC3 cell line after overexpression of PR to show that progesterone decreases transcription of the AR gene by binding to the 5’UTR of the AR gene. As a mechanism of action, the authors showed that PR decreased acetylation in H3 of the 5’UTR, while an inhibitor of the class I HDAC enzyme restored normal AR expression. Lastly, through a ChIP-loop assay, the authors showed that AR and PR can bind both 5’UTR and the second intron region simultaneously by generating a loop in the DNA explaining the importance of the second intron binding site for AR and the activity of PR in this system.

In general, this work supports a novel mechanism of AR regulation, which might be pertinent to different human pathologies. Despite this, some additional experiments are needed to strengthen these results.

Major revisions:

1. The authors should further demonstrate that the observed effect is through P4-PR interactions. Some prostate cancer cells are known to perform neo steroidogenesis from cholesterol or progesterone precursors, is there any analog that can be used instead of P4? Additionally, some progesterone analogs have been shown to activate the AR. The use of specific antagonists for PR are recommended.

2. Since prostate cancer cell lines are used in this study, the authors should provide evidence for co-expression of AR and PR, at least at the mRNA level in publicly available prostate cancer patient datasets and for protein in a panel of cell lines.

3. Was any effect of P4 observed in untransfected cells?

4. It is unclear why CFP antibody stains the lysates from untransfected VCaP cells in figure S2C.

5. As stated by the authors, VCaP cells express variants of the AR, what are the effects of PR on AR variants? These might be relevant in cases of CRPC.

6. A dose response curve of progesterone showing effect on AR or its mRNA should be conducted.

7. The authors observed a decrease in AR expression in VCaP cells after DHT treatment. Several authors have shown in different prostate cancer cell lines that AR can decrease transcriptional activity of its own gene, yet AR protein is stabilized by ligand binding leading to an increase in AR protein. The authors should discuss this discrepancy in their findings vs the published literature.

8. Considering their ChIP-qPCR experiments, the authors argue that AR and PR bind preferentially to the intron 2 compared to the 5’UTR region of the AR gene. This interpretation could be incorrect as there may be differences intrinsic to the experiment such as variation in primer binding efficiency.

9. Authors show that acetylation of histone 3 in the 5’UTR of the AR gene is a potential mechanism of action explaining effects of PR. Additionally the authors assess the role of HDAC using an inhibitor of HDAC class I. As any pharmacological inhibitor, off-target effects of sodium butyrate have been reported (Kruh J Mol Cell Biochem 1982), authors should discuss the limitations of this assay.

10. Ectopic expression of cDNAs can lead to very high levels of protein, authors should consider analyzing a cell line that has endogenous expression of AR and PR.

11. A major concern of this study is the statistical analysis performed. Western blot experiments seem to be normalized to vehicle treated cells, this leads to a lack of homogeneity of the variance which is one of the assumptions of the t-tests. Authors should control this and use the proper statistical analysis.

Minor revisions:

12. Distal AR super enhancers have been demonstrated in several publications. The authors should discuss this point and how it might impact interpretation of their work.

6. PLOS authors have the option to publish the peer review history of their article (what does this mean?). If published, this will include your full peer review and any attached files.

Reviewer #1: No

Reviewer #2: No

---

## [Author Response · Author response to Decision Letter 0]

4 Dec 2023

15th August 2023

Lucia R Languino, PhD

Academic Editor

PLOS ONE

RE: PONE-D-23-27820

Title: Mechanistic Insights into Steroid Hormone-mediated Regulation of the Androgen Receptor Gene

Dear Dr Languino,

Thank you for your e-mail of September 18th and the Reviewers for their constructive criticisms and suggestions. We would now like to submit our revised manuscript and have detailed below our careful consideration of the points raised and the changes we have made as a consequence of the Reviewers’ concerns. We have also included in the revised manuscript copies of uncropped gel images and western blots as supplementary figures.

We believe the changes made have helped clarify our analysis and have strengthened the overall rigor and conclusions of our study. 

Please note, as the communicating author and a member of staff at the University of Aberdeen: ‘For the purpose of open access, the author has applied a Creative Commons Attribution (CC BY) [or other appropriate open licence] licence to any Author Accepted Manuscript version arising from this submission’.

Best Regards,

Iain J. McEwan, PhD

Professor in Molecular and Cellular Endocrinology

Research Lead for Institute of Medical Sciences

e-mail: iain.mcewan@abdn.ac.uk

Response to Reviewers

Reviewer 1

1. We agree with the reviewer that it is important to consider the impact of exogenous expression of proteins. However, it is important to note that all the experiments reported with exogenous expression of the progesterone receptor (PR) and endogenous levels of androgen receptor (AR) are plus/ minus hormone. In the ChIP experiments (Figure 3) for example there is clear recruitment of AR and PR after hormone treatment which is not seen with a non-specific IgG control. We also note that VCaP cells are widely reported to over express the endogenous AR and a comparison of the levels of AR (Figure 2, Supplementary Figure S1) with exogenous PR (A/B) (Supplementary Figure S2) are not so different, relative to the house keeping control. In contrast we do see significant over expression of the ERα (Supplementary Figure S2), but no significant impact of this receptor in the absence or presence of hormone. Furthermore, we have redrawn the data in Supplementary Figure S4 showing that there is no effect of P4 in the absence of PR. Therefore, we conclude our results cannot simply be explained because of receptor overexpression and collectively, we can be confident that we are seeing effects of the hormone activated PRB in our experiments.

2. We apologise for this oversight, the number of independent blots was n=3, which is now included in the figure legend. We now also include the full blots as supplementary material (Supplementary Figure S11).

3. We thank the reviewer for this comment and would like to take this opportunity to state that our experiments were designed and analysed carefully and apologise that this did not come across in the text. We have now corrected the text throughout for clarity and to demonstrate careful analysis and interpretation of the results as suggested. 

4. As point 3 above. 

5. As point 3 above. 

6. Agreed and this has been done.

7. We agree this was an oversight and we have now included the additional references [13 to 16] to a far upstream enhancer element. However, this needs to be placed in the context of the present study which was addressing the mechanism of steroid receptor downregulation of the AR mRNA. The upstream enhancer is interesting as it appears to have features of a switch regulating AR during development and reactivated during progression to therapy resistant prostate cancer. However, there is no evidence it is involved in the AR (our work and others) or PR-negative regulation (present study) of receptor mRNA. We have also revised the text of this paragraph to more accurately reflect the work cited (lines 85 to 101).

8. We are happy to acknowledge limitations of the present study (Discussion). However, the role of PR in the regulation of the AR gene is supported by other studies in the literature and our own analysis of publicly available data sets. Our work contributes to this body of evidence by providing mechanistic insights, which can be further tested, for example in a cell model with endogenous levels of both AR and PR (line788-789).

9. Corrected

10. The text has been proofread and corrected where necessary.

11. Text has been corrected as suggested.

Reviewer 2

1. We agree with the reviewer this is an important point and indeed informed our choice of VCaP and PC3 (AR negative) cells as our principal cell models. LNCaP cells which are widely used due to being hormone-sensitivity and ease of growing, have mutated AR (T878A), which permits activation by progesterone amongst other ligands. Similarly, 22RV1 cells have mutated AR. To address this concern, we have revised Supplementary Figure 4, to indicate that there was no effect of P4 alone in the absence of PR transfection, eliminating the possible effects of P4 on AR. Unfortunately, we did not have a PR antagonist in these studies.

2. We thank the reviewer for this comment. We have analysed patient data as suggested- included here for review purposes only. We are planning a separate manuscript focusing on the role of PR in prostate cancer, including our own IHC data and patient data sets. However as shown in the figure below (TCGA data set), there is a clear decrease in PR levels, which contrast with increased AR expression and is consistent with our model that loss of PR repression will lead to increased levels of AR.

See Reply to Reviewers document for Figure

Figure Legend- Expression of Steroid Hormone Receptor genes in primary prostate tumours. Expression of A) AR; B) PGR; C) ESR1 were analysed in the 333 primary prostate cancer samples available through the TCGA Research Network (https://www.cancer.gov/tcga). This analysis was carried out using PIXdb (Marzec et al., 2021). Relative expression of these genes in normal tissue (red) and prostatic tumours (blue) are shown compared to the z-score for that RNA. Z-score of a gene is calculated by comparing its expression level in a given sample to the expression level of that gene across all samples.

Our aim with the present study- manuscript - was to investigate the underpinning mechanism of steroid receptor regulation of the AR gene.

3. No there was no effect of P4 in untransfected cells, and we have clarified this with revision of Supplementary Figure S4 (see also comments to Point 1 above).

4. We agree with the reviewer this is puzzling. The assumption is cross-reactivity of the antibody and have made a comment to this effect.

5. Although this manuscript was not focussing on prostate cancer specifically, we agree that the regulation of splice variants is a very important and an interesting aspect of AR biology. We have now included data for both 22RV1 and VCaP cells showing hormone (DHT or P4) regulation of AR-Vs (new Supplementary Figure S5) (text lines 409 to 417). 

6. We agree this would be interesting to include, but a single concentration of hormone was chosen for consistency across the multiple experimental methods used.

7. In our VCaP cells, change in AR RNA correlated with changes in receptor protein-levels, with DHT treatment significantly reducing both transcript and protein levels. Whilst DHT treatment reduced AR transcript to only one quarter of control levels, the effect on AR protein was much less pronounced, with less than a 50% reduction noted. This has been reported widely previously by us and others: while DHT-mediated repression of the AR protein has also been noted in VCaP cells previously. However, as stated by the reviewer, investigation in other prostate cancer cell lines have noted DHT-mediated increase in total AR protein in LNCaP, 22Rv1 and LAPC4 cells (Cai et al., 2011; Krongrad et al., 1991). The reason for this difference is unclear but maybe related to a difference in baseline AR expression: VCaP cells contain an amplification of the AR gene, in fact possessing almost thirty copies of the AR gene, resulting in vastly increased AR RNA and protein levels (Korenchuk et al., 2001; Makkonen and Palvimo, 2011). As such, it is possible that the effect of DHT on AR stabilisation is somewhat diluted versus the total pool of available AR, and so the effects of reduced protein production can be noted. Thus, in LNCaP androgen reduces mRNA but stabilises protein, while in VCaP androgens decrease both mRNA and protein. To the best of our knowledge the mechanistic basis for this is unknown. However, we have added a sentence to clarify this in the text (lines 424-427).

8. We appreciate the concern raised and have added a sentence to highlight this point (lines 693-695).

9. Again, we appreciate the concern raised and include additional supporting data to show that the concentrations of SAHA or sodium butyrate used do not impact on control AR mRNA (Supplementary Figure S10).

10. Due to mutations in the AR leading to promiscuous hormone binding we were limited in the choice of prostate cell models. The reviewer makes a good suggestion, and a follow-up study might address this in breast cancer model (MCF-7) where both AR and PR are endogenously expressed. As discussed above neither the over expression of PR alone or P4 alone is sufficient to regulate AR gene expression. However, we have now included a statement regarding the value of repeating the study with cells expressing AR and PR endogenously (lines 788-789).

11. This is an important point and we have now clarified the analysis and statistical tests used throughout in the revised manuscript and updated the relevant part of the methods.

12. We agree and have expanded the text of the Introduction to acknowledge this work on a distal enhancer and provide a fuller description of previous work [references 13 to 16] (lines 85 to 101).

---

## [Decision Letter · Decision Letter 1]

18 Dec 2023

PONE-D-23-27820R1Mechanistic Insights into Steroid Hormone-mediated Regulation of the Androgen Receptor GenePLOS ONE

Dear Dr. McEwan,

Thank you for submitting your revised manuscript to PLOS ONE. After careful consideration, we feel that it has merit but does not fully meet PLOS ONE’s publication criteria as it currently stands. Therefore, we invite you to submit a re-revised version of the manuscript that addresses the points raised during the review process.

Some of the new data show inconsistencies and more importantly key questions have not been answered.

Alll new comments need to be addressed in a re-revised version of the manuscript.

We look forward to receiving your revised manuscript.

Kind regards,

Lucia R. Languino, Ph.D.

Academic Editor

PLOS ONE

Additional Editor Comments (if provided):

Alll new comments need to be addressed.

Reviewers' comments:

Reviewer's Responses to Questions

**Comments to the Author**

1. If the authors have adequately addressed your comments raised in a previous round of review and you feel that this manuscript is now acceptable for publication, you may indicate that here to bypass the “Comments to the Author” section, enter your conflict of interest statement in the “Confidential to Editor” section, and submit your "Accept" recommendation.

Reviewer #1: (No Response)

Reviewer #2: (No Response)

2. Is the manuscript technically sound, and do the data support the conclusions?

Reviewer #1: Yes

Reviewer #2: Partly

3. Has the statistical analysis been performed appropriately and rigorously? 

Reviewer #1: Yes

Reviewer #2: No

4. Have the authors made all data underlying the findings in their manuscript fully available?

Reviewer #1: Yes

Reviewer #2: Yes

5. Is the manuscript presented in an intelligible fashion and written in standard English?

Reviewer #1: Yes

Reviewer #2: Yes

6. Review Comments to the Author

Reviewer #1: There are a few minor typos in the new text of the revision that should be corrected. A simple spell-check should identify these (for instance, lines 409-417 of the add/track manuscript).

Reviewer #2: The authors have answered several of my concerns and added some new interesting data to their manuscript as well as supplementary information. However, some of the new data shows inconsistencies and more importantly key questions have not been answered.

Specific comments:

Overexpression of proteins can lead to non-physiological effects. In this particular case, it is known that AR and PR share a high degree of homology in their DNA-binding domain (Zhifeng Zhou et al. JBC Volume 272, Issue 13, 28 March 1997, Pages 8227-8235). All experiments were done using an overexpression approach thus, a main concern is if the effect observed in this work is physiological. Using cells that endogenously express both AR and PR would resolve this issue.

1. As for the following comment from my previous review of the MS:

“Considering their ChIP-qPCR experiments, the authors argue that AR and PR bind preferentially to the intron 2 compared to the 5’UTR region of the AR gene. This interpretation could be incorrect as there may be differences intrinsic to the experiment such as variation in primer binding efficiency.”

Authors have added a sentence regarding the possibility of differential primer binding efficiency, yet authors still mistakenly interpret their results stating that that the intron 2 response element is the preferential binding site for hormone.

2. Decrease in AR expression in PR-B cells after P4 stimulus is not evident in figure 2B. qPCR results are a good complement for corroboration of this result. Yet primer sequences for PR-A and PR-B as well as for the ERs are not specified in supplementary table 2, impeding the possibility of the reader to corroborate specificity. In this regard, due to the importance of this qPCR for the conclusions of the MS, it would be necessary for authors to confirm that primers are amplifying only one band at the correct size and ideally sequencing that band to demonstrate specificity.

3. In addition to my previous comment on figure 2B. Supplementary figure S11 does not correlate to Figure 2 making it impossible to determine effects by WB. B-actin and PSA from the right panel of figure 2A (PR-A and PR-B) seem to belong to blot 1 but lanes are from PR-B and ER-a. In the case of AR, the PR-A and PR-B seem to belong to PR-A, Dex, DHT and Vehicle.

4. Additionally, it is suggested that authors look at well-known downstream AR-targets such as KLK2, KLK3, TMPRSS2, NKX3-1, FKBP5 in cells stimulated with P4. This would add considerable strength to the hypothesis.

7. PLOS authors have the option to publish the peer review history of their article (what does this mean?). If published, this will include your full peer review and any attached files.

Reviewer #1: No

Reviewer #2: No

---

## [Author Response · Author response to Decision Letter 1]

24 Jan 2024

Please see Cover Letter to Editor and accompanying word document.

---

## [Decision Letter · Decision Letter 2]

18 Feb 2024

PONE-D-23-27820R2Mechanistic Insights into Steroid Hormone-mediated Regulation of the Androgen Receptor GenePLOS ONE

Dear Dr. McEwan,

Thank you for submitting your manuscript to PLOS ONE. After careful consideration, we feel that it has merit but does not fully meet PLOS ONE’s publication criteria as it currently stands. Therefore, we invite you to submit a re-revised version of the manuscript that addresses the points raised during the review process.  It is felt thta data supporting the relevance of your cell models are tsilll missing.

Please submit your re-revised manuscript by Apr 03 2024 11:59PM. If you will need more time than this to complete your revisions, please reply to this message or contact the journal office at plosone@plos.org. Please include the following items when submitting your revised manuscript:A rebuttal letter that responds to each point raised by the academic editor and reviewer(s). You should upload this letter as a separate file labeled 'Response to Reviewers'.A marked-up copy of your manuscript that highlights changes made to the original version. You should upload this as a separate file labeled 'Revised Manuscript with Track Changes'.An unmarked version of your revised paper without tracked changes. You should upload this as a separate file labeled 'Manuscript'.

We look forward to receiving your revised manuscript.

Kind regards,

Lucia R. Languino, Ph.D.

Academic Editor

PLOS ONE

Reviewers' comments:

Reviewer's Responses to Questions

**Comments to the Author**

1. If the authors have adequately addressed your comments raised in a previous round of review and you feel that this manuscript is now acceptable for publication, you may indicate that here to bypass the “Comments to the Author” section, enter your conflict of interest statement in the “Confidential to Editor” section, and submit your "Accept" recommendation.

Reviewer #2: (No Response)

2. Is the manuscript technically sound, and do the data support the conclusions?

Reviewer #2: Partly

3. Has the statistical analysis been performed appropriately and rigorously? 

Reviewer #2: Yes

4. Have the authors made all data underlying the findings in their manuscript fully available?

Reviewer #2: Yes

5. Is the manuscript presented in an intelligible fashion and written in standard English?

Reviewer #2: Yes

6. Review Comments to the Author

Reviewer #2: The authors have not adequately addressed my concerns including those relating to the relevance of the overexpression model system. Also, comparable efficiency of the primer sets remains questionable.

7. PLOS authors have the option to publish the peer review history of their article (what does this mean?). If published, this will include your full peer review and any attached files.

Reviewer #2: No

---

## [Author Response · Author response to Decision Letter 2]

10 Apr 2024

Response to Reviewers-R1

Reviewer 1

1. We agree with the reviewer that it is important to consider the impact of exogenous expression of proteins. However, it is important to note that all the experiments reported with exogenous expression of the progesterone receptor (PR) and endogenous levels of androgen receptor (AR) are plus/ minus hormone. In the ChIP experiments (Figure 3) for example there is clear recruitment of AR and PR after hormone treatment which is not seen with a non-specific IgG control. We also note that VCaP cells are widely reported to over express the endogenous AR and a comparison of the levels of AR (Figure 2, Supplementary Figure S1) with exogenous PR (A/B) (Supplementary Figure S2) are not so different, relative to the house keeping control. In contrast we do see significant over expression of the ERα (Supplementary Figure S2), but no significant impact of this receptor in the absence or presence of hormone. Furthermore, we have redrawn the data in Supplementary Figure S4 showing that there is no effect of P4 in the absence of PR. Therefore, we conclude our results cannot simply be explained because of receptor overexpression and collectively, we can be confident that we are seeing effects of the hormone activated PRB in our experiments.

2. We apologise for this oversight, the number of independent blots was n=3, which is now included in the figure legend. We now also include the full blots as supplementary material (Supplementary Figure S11).

3. We thank the reviewer for this comment and would like to take this opportunity to state that our experiments were designed and analysed carefully and apologise that this did not come across in the text. We have now corrected the text throughout for clarity and to demonstrate careful analysis and interpretation of the results as suggested. 

4. As point 3 above. 

5. As point 3 above. 

6. Agreed and this has been done.

7. We agree this was an oversight and we have now included the additional references [13 to 16] to a far upstream enhancer element. However, this needs to be placed in the context of the present study which was addressing the mechanism of steroid receptor downregulation of the AR mRNA. The upstream enhancer is interesting as it appears to have features of a switch regulating AR during development and reactivated during progression to therapy resistant prostate cancer. However, there is no evidence it is involved in the AR (our work and others) or PR-negative regulation (present study) of receptor mRNA. We have also revised the text of this paragraph to more accurately reflect the work cited (lines 85 to 101).

8. We are happy to acknowledge limitations of the present study (Discussion). However, the role of PR in the regulation of the AR gene is supported by other studies in the literature and our own analysis of publicly available data sets. Our work contributes to this body of evidence by providing mechanistic insights, which can be further tested, for example in a cell model with endogenous levels of both AR and PR (line788-789).

9. Corrected

10. The text has been proofread and corrected where necessary.

11. Text has been corrected as suggested.

Reviewer 2

1. We agree with the reviewer this is an important point and indeed informed our choice of VCaP and PC3 (AR negative) cells as our principal cell models. LNCaP cells which are widely used due to being hormone-sensitivity and ease of growing, have mutated AR (T878A), which permits activation by progesterone amongst other ligands. Similarly, 22RV1 cells have mutated AR. To address this concern, we have revised Supplementary Figure 4, to indicate that there was no effect of P4 alone in the absence of PR transfection, eliminating the possible effects of P4 on AR. Unfortunately, we did not have a PR antagonist in these studies.

2. We thank the reviewer for this comment. We have analysed patient data as suggested- included here for review purposes only. We are planning a separate manuscript focusing on the role of PR in prostate cancer, including our own IHC data and patient data sets. However as shown in the figure below (TCGA data set), there is a clear decrease in PR levels, which contrast with increased AR expression and is consistent with our model that loss of PR repression will lead to increased levels of AR.

Our prim

FFigure Legend- Expression of Steroid Hormone Receptor genes in primary prostate tumours. Expression of A) AR; B) PGR; C) ESR1 were analysed in the 333 primary prostate cancer samples available through the TCGA Research Network (https://www.cancer.gov/tcga). This analysis was carried out using PIXdb (Marzec et al., 2021). Relative expression of these genes in normal tissue (red) and prostatic tumours (blue) are shown compared to the z-score for that RNA. Z-score of a gene is calculated by comparing its expression level in a given sample to the expression level of that gene across all samples.

Our aim with the present study- manuscript - was to investigate the underpinning mechanism of steroid receptor regulation of the AR gene.

3. No there was no effect of P4 in untransfected cells, and we have clarified this with revision of Supplementary Figure S4 (see also comments to Point 1 above).

4. We agree with the reviewer this is puzzling. The assumption is cross-reactivity of the antibody and have made a comment to this effect.

5. Although this manuscript was not focussing on prostate cancer specifically, we agree that the regulation of splice variants is a very important and an interesting aspect of AR biology. We have now included data for both 22RV1 and VCaP cells showing hormone (DHT or P4) regulation of AR-Vs (new Supplementary Figure S5) (text lines 409 to 417). 

6. We agree this would be interesting to include, but a single concentration of hormone was chosen for consistency across the multiple experimental methods used.

7. In our VCaP cells, change in AR RNA correlated with changes in receptor protein-levels, with DHT treatment significantly reducing both transcript and protein levels. Whilst DHT treatment reduced AR transcript to only one quarter of control levels, the effect on AR protein was much less pronounced, with less than a 50% reduction noted. This has been reported widely previously by us and others: while DHT-mediated repression of the AR protein has also been noted in VCaP cells previously. However, as stated by the reviewer, investigation in other prostate cancer cell lines have noted DHT-mediated increase in total AR protein in LNCaP, 22Rv1 and LAPC4 cells (Cai et al., 2011; Krongrad et al., 1991). The reason for this difference is unclear but maybe related to a difference in baseline AR expression: VCaP cells contain an amplification of the AR gene, in fact possessing almost thirty copies of the AR gene, resulting in vastly increased AR RNA and protein levels (Korenchuk et al., 2001; Makkonen and Palvimo, 2011). As such, it is possible that the effect of DHT on AR stabilisation is somewhat diluted versus the total pool of available AR, and so the effects of reduced protein production can be noted. Thus, in LNCaP androgen reduces mRNA but stabilises protein, while in VCaP androgens decrease both mRNA and protein. To the best of our knowledge the mechanistic basis for this is unknown. However, we have added a sentence to clarify this in the text (lines 424-427).

8. We appreciate the concern raised and have added a sentence to highlight this point (lines 693-695).

9. Again, we appreciate the concern raised and include additional supporting data to show that the concentrations of SAHA or sodium butyrate used do not impact on control AR mRNA (Supplementary Figure S10).

10. Due to mutations in the AR leading to promiscuous hormone binding we were limited in the choice of prostate cell models. The reviewer makes a good suggestion, and a follow-up study might address this in breast cancer model (MCF-7) where both AR and PR are endogenously expressed. As discussed above neither the over expression of PR alone or P4 alone is sufficient to regulate AR gene expression. However, we have now included a statement regarding the value of repeating the study with cells expressing AR and PR endogenously (lines 788-789).

11. This is an important point and we have now clarified the analysis and statistical tests used throughout in the revised manuscript and updated the relevant part of the methods.

12. We agree and have expanded the text of the Introduction to acknowledge this work on a distal enhancer and provide a fuller description of previous work [references 13 to 16] (lines 85 to 101).

Response to Reviewers’ Comments R2

Reviewer 1

We have carefully read over the revised manuscript and have corrected all typos found.

Reviewer 2

We thank the reviewer for their additional comments and suggestions.

As we explained in out earlier response to the concern on the use exogenous expression of the PR, there is clear methodological rationale and the effects observed are regulated by hormone, not simply the presence of the receptor protein. We evidence this by the data in Supplementary Figure S4 and throughout the manuscript. We also acknowledged the limitations of this approach in our earlier revision of the manuscript and the importance of further studies in cells expressing endogenous AR and PR. However, these experiments are out with the scope of the present study. However, it is also important to stress the use of exogenous expression is both a widely used and accepted methodology in dissecting molecular signalling pathways.

It is precisely because the 3-keto receptors (AR, GR, PR and MR) share homology in their DNA binding domains and in the DNA recognition sequences of hormone response elements that we were excited to report the detailed molecular programme of work described in this manuscript. Indeed, we reference this point in the Introduction (text lines 108 to 114).

1. Interpreting the ChIP studies. We have now included additional supporting data demonstrating that the primer sets target the two genomic regions of the AR gene with comparable efficiencies (Supplementary Figure S7, text lines 451 to 452). Therefore, collectively we would argue that our interpretation of the evidence presented is consistent with preferential recruitment of both the AR and PR-B at the Intronic site. We accept that other interpretations maybe possible, that is the nature of science. 

2. RT-qPCR was not done on either the ER or PR mRNA transcripts, which is why there is no information on primers. All primers used in qPCR are listed in Table 2 and we have now included a Supplementary Figure (S5) showing the specificity of the AR primers used and a single amplified product of 103 bp (text lines 377 to 382).

3. We are grateful to the reviewer for drawing our attention to this error. We have now included the correct gel images in Figure 2A (middle and right panels). It is important to state this does not affect the quantification shown in Part B and subsequent conclusions. 

4. This is an interesting point; however, the main conclusions of this study were the regulation of the AR gene by progesterone acting through PR-B protein. In this context it should be noted that we do see an impact on the AR target gene PSA (KLK3) protein levels in response to P4/ PR-B, most likely as a consequence of downregulated AR expression (Figure 2C and text lines 419 to 421).

---

## [Editor Report · Decision Letter 3]

8 May 2024

Mechanistic Insights into Steroid Hormone-mediated Regulation of the Androgen Receptor Gene

PONE-D-23-27820R3

Dear Dr. McEwan,

We’re pleased to inform you that your manuscript has been judged scientifically suitable for publication and will be formally accepted for publication once it meets all outstanding technical requirements.

Kind regards,

Zoran Culig

Academic Editor

PLOS ONE

Additional Editor Comments (optional):

No further comments necessary.
---

## [Editor Report · Acceptance letter]

27 Jun 2024

PONE-D-23-27820R3 

PLOS ONE

Dear Dr. McEwan, 

I'm pleased to inform you that your manuscript has been deemed suitable for publication in PLOS ONE. Congratulations! Your manuscript is now being handed over to our production team.

Kind regards, 

on behalf of

Dr. Zoran Culig 

Academic Editor

PLOS ONE